# Natural variation in *BnaA07.MKK9* confers resistance to Sclerotinia stem rot in oilseed rape

Li Lin [1,2], Xingrui Zhang [1], Jialin Fan[1], Jiawei Li [3], Sichao Ren [1], Xin Gu [1], Panpan Li[1], Meiling Xu [2], Jingyi Xu [1], Wenjing Lei [1], Dongxiao Liu [2], Qinfu Sun [2], Guangqin Cai[4], Qing-Yong Yang [3], Youping Wang [1,2] ✉ & Jian Wu [1] ✉

Sclerotinia stem rot (SSR), caused by the necrotrophic fungus *Sclerotinia sclerotiorum*, is one of the most devastating diseases for several major oil-producing crops. Despite its impact, the genetic basis of SSR resistance in plants remains poorly understood. Here, through a genome-wide association study, we identify a key gene, *BnaA07. MKK9*, that encodes a mitogen-activated protein kinase kinase that confers SSR resistance in oilseed rape. Our functional analyses reveal that BnaA07.MKK9 interacts with BnaC03.MPK3 and BnaC03.MPK6 and phosphorylates them at the TEY activation motif, triggering a signaling cascade that initiates biosynthesis of ethylene, camalexin, and indole glucosinolates, and promotes accumulation of $H_2O_2$ and the hypersensitive response, ultimately conferring resistance. Furthermore, variations in the coding sequence of *BnaA07.MKK9* alter its kinase activity and improve SSR resistance by ~30% in cultivars carrying the advantageous haplotype. These findings enhance our understanding of SSR resistance and may help engineer novel diversity for future breeding of oilseed rape.

Oilseed rape (*Brassica napus* L. AACC, 2*n* = 38), also known as canola or rapeseed, is a globally important edible oil crop that emerged during the past 7500 years from natural hybridization between *B. rapa* (AA, 2*n* = 20) and *B. oleracea* (CC, 2*n* = 18)[1]. Despite a relatively short history of domestication, oilseed rape has undergone significant development. The original winter type was first cultivated in 13th century Europe, while the spring-type dates back to around 1700[2]. After its introduction to China in the 1930s–1940s, intensive selection led to the development and widespread cultivation of the semi-winter-type *B. napus* in China[3].

Over the past five decades, double-low cultivars with low seed erucic acid and glucosinolate content, pioneered in Canada, significantly improved *B. napus* oil and meal quality. This program focused on lowering seed glucosinolates and erucic-acid levels to improve seed quality, making it the most noteworthy event in the breeding history of oilseed rape.

Meanwhile, Sclerotinia stem rot (SSR), caused by the necrotrophic fungus *Sclerotinia sclerotiorum*, is a widespread, destructive disease affecting oilseed rape. It leads to significant economic losses in oilseed rape-growing regions, including China and Canada. In Canada, the annual average incidence of SSR in oilseed rape ranges from 10–20%[4], but in 2016, it reached over 90% of fields in Western Canada[5]. In China, yield losses caused by SSR range from 10–20% annually, with up to 80% losses in severely infected fields[6]. While fungicide application is the

[1]Key Laboratory of Plant Functional Genomics of the Ministry of Education, Yangzhou University, Yangzhou 225009, China. [2]Jiangsu Key Laboratory of Crop Genomics and Molecular Breeding, Yangzhou University, Yangzhou 225009, China. [3]Hubei Key Laboratory of Agricultural Bioinformatics, College of Informatics, Huazhong Agricultural University, Wuhan 430070, China. [4]Key Laboratory of Biology and Genetic Improvement of Oil Crops, Ministry of Agriculture and Rural Affairs, Oil Crop Research Institute, Chinese Academy of Agricultural Sciences, Wuhan 430070, China. ✉e-mail: wangyp@yzu.edu.cn; wu_jian@yzu.edu.cn

most common management strategy, determining the optimal time to spray is challenging, often leading to ineffective treatment and environmental contamination. Therefore, breeding resistant oilseed rape varieties is considered the most effective and sustainable approach to combat SSR.

Since no oilseed rape germplasm, nor any of its close relatives is completely immune to SSR, breeding for resistance in *B. napus* is heavily reliant on selecting germplasms with genes that confer partial resistance. SSR resistance is controlled by multiple quantitative trait loci (QTLs), making it a complex trait to enhance through breeding[7,8]. While bi-parental mapping has identified dozens of SSR resistance-associated QTLs in *B. napus*[9], none have been fine-mapped and cloned, limiting their utility in breeding. Despite advances in high-throughput genotyping technologies and the availability detailed reference genomes, SSR resistance genes in oilseed rape remain elusive.

With the advent of genome-wide association studies (GWAS), which leverage detailed genetic information and advanced genotyping, researchers have begun to dissect the complex genetics of disease resistance in various crops. In recent years, GWAS have identified candidate resistance genes for rice sheath blight[10], rice blast disease[11,12], and cotton verticillium wilt[13,14], among others. Despite the many GWAS investigating SSR resistance in oilseed rape, no specific genes related to SSR resistance have been conclusively characterized to date[7,15-17].

To combat pathogen infection, plants have evolved two main innate immune systems: effector-triggered immunity (ETI) and PAMP-triggered immunity (PTI)[18]. ETI, mediated by resistance (*R*) genes, is effective against biotrophic and hemibiotrophic pathogens, but less so against *S. sclerotiorum* and other necrotrophic pathogens. PTI, a form of quantitative resistance, provides broader protection and is vital in the defense against necrotrophic pathogens[19,20].

Central to PTI is the mitogen-activated protein kinase (MAPK) cascade, involving a sequence of phosphorylation events crucial for signal transduction[21]. MAP kinase kinases (MKKs), a small family within the MAPK cascade, are key players. Once MKKs are activated by MKKKs, they continue the cascade by activating MAPKs[22]. Several MKKs have been well-documented as important for plant immunity with varied roles ranging from positive to negative regulation of immune responses[23-25].

In this study, we identify the MKK gene *BnaA07.MKK9* as a significant factor contributing to SSR resistance in oilseed rape through GWAS analysis. Working upstream of *BnaC03.MPK3/6*, *BnaA07.MKK9* initiates biosynthesis of defense-related compounds, including ethylene (ET), camalexin, and indole glucosinolates (IGSs), and promoted $H_2O_2$ accumulation in response to *S. sclerotiorum* infection, ultimately conferring resistance. Variations in *BnaA07.MKK9*, influenced by geography, ecotype, and breeding period, correspond to differences in kinase activity, and, consequently, in the disease resistance response. This discovery highlights the importance of *BnaA07.MKK9* in SSR resistance and enhances our understanding of the genetic and molecular basis underlying SSR resistance in oilseed rape.

## Results

### *BnaA07.MKK9* is associated with SSR resistance in oilseed rape
SSR resistance in oilseed rape is a complex, quantitative inherited trait that is determined by multiple genes[16,17]. To pinpoint the genes involved, we evaluated SSR resistance in a group of 322 oilseed rape accessions using a stem inoculation assay at the mature stage in 2016. While none of these accessions showed complete immunity, the severity of SSR varied significantly, with stem lesions ranging from 1.3 to 11.6 cm at 168 h post-inoculation (hpi) (Fig. 1a). The broad-sense heritability ($h^2$) for SSR resistance was estimated to be 72%, suggesting that genetic variance predominantly contributes to the observed phenotypic variations.

A total of 4,984,924 single-nucleotide polymorphisms (SNPs), generated from whole genome re-sequencing of this oilseed rape

panel, were collected from the BnIR database (https://yanglab.hzau.edu.cn/BnIR). A phylogenetic analysis based on SNPs elucidated the relationships among 322 accessions, revealing their classification into two major sub-populations, designated as G1 and G2 (Supplementary Fig. 1a). The G2 sub-population comprised 288 accessions; while the G1 sub-population comprised 34, with 26 of them being spring-type ecotypes (Supplementary Data 1). Principal component analysis (PCA) revealed that the G1 and G2 sub-populations formed distinct clusters, consistent with the findings from the structure analysis (Supplementary Fig. 1b). GWAS performed using a mixed linear model uncovered 126 SNPs ($P < 2 \times 10^{-7}$ (1/4,984,924)) associated with SSR resistance (Fig. 1b, c), all of which were located between 24.439 and 24.590 Mb on chromosome A7 of the ZS11 reference genome, with a notable cluster of 120 SNPs within a 21 kb interval (24.499–24.520 Mb). This region contained six protein-coding genes (Fig. 1d).

The gene BnaA07G0255500ZS within this region emerged as a key candidate for SSR resistance due to its significant up-regulation in both leaves and stems following *S. sclerotiorum* inoculation (Fig. 1e). Furthermore, the gene contained three of the leading SNPs within an unambiguous linkage disequilibrium block (Fig. 1d). BnaA07G0255500ZS (*BnaA07.MKK9*) is homologous to *Arabidopsis Mitogen-activated protein kinase kinase 9* (*AtMKK9*), a gene that affects various physiological processes, such as salt tolerance[26], leaf senescence[27], and phosphate acquisition[28]. However, its role in the plant immune response remains unclear.

To gain additional insights into its role, we examined changes associated with this gene at both the protein and kinase activity levels following inoculation with *S. sclerotiorum*. There was a significant increase in both the protein abundance (Fig. 1f) and kinase activity (Fig. 1g) of BnaA07.MKK9 upon *S. sclerotiorum* infection in oilseed rape leaves, further supporting a role for *BnaA07.MKK9* in the immune response of oilseed rape to SSR.

### *BnaA07.MKK9* positively regulated resistance against *S. sclerotiorum*
To investigate how *BnaA07.MKK9* responds to *S. sclerotiorum*, we generated transgenic oilseed rape lines that overexpress (OE) *BnaA07.MKK9^{DD}*, a kinase-active variant of *BnaA07.MKK9* with active loop residues Thr[192] and Ser[198] mutated to Asp. These modifications were selected based on data from a previous *Arabidopsis* study[26]. The *BnaA07.MKK9^{DD}* gene, under control of the CaMV 35S promoter, was transformed into the pure oilseed rape line, J9712, via *Agrobacterium tumefaciens*-mediated transformation (Supplementary Fig. 2a). Presence of the OE gene in transformed lines (T_0 progeny) was confirmed with reverse transcription PCR (RT-PCR) (Supplementary Fig. 2b). We then evaluated SSR resistance in three independent T_1 generation *BnaA07.MKK9^{DD}*-OE plants (#12, #35, and #51), alongside the non-transgenic J9712 line and plants transformed with an empty vector (EV). Compared to the J9712 line, transgenic OE plants had 12.4–22.8% smaller lesions on leaves at 48 h post-inoculation (hpi) (Fig. 2a, b) and 15.6–31.9% shorter lesions on stems at 168 hpi (Fig. 2c, d). These improvements were consistent in T_2 progenies (Supplementary Fig. 2c–f).

In parallel, CRISPR/Cas9-mediated genome editing was used to generate *Bnamkk9* knockout mutants in oilseed rape, targeting all four of the *BnaMKK9* homologs in the ZS11 reference genome. These homologs, *BnaA07.MKK9* (BnaA07G0255500ZS), *BnaC06.MKK9* (BnaC06G0283400ZS), *BnaA02.MKK9* (BnaA02G0206900ZS) and *BnaC02.MKK9* (BnaC02G0277100ZS), exhibited high nucleotide and protein similarity (Supplementary Fig. 3a), and three out of four exhibited strong induction in both leaves and stems after inoculation with *S. sclerotiorum* (Supplementary Fig. 3b). Thus, four gRNAs were designed for editing of *BnaMKK9* (Supplementary Fig. 3c, d). Out of 11 T_0-positive transgenic plants, two (#5 and #7) contained simultaneous editing events at *BnaA07.MKK9*, *BnaC06.MKK9* and *BnaA02.MKK9*

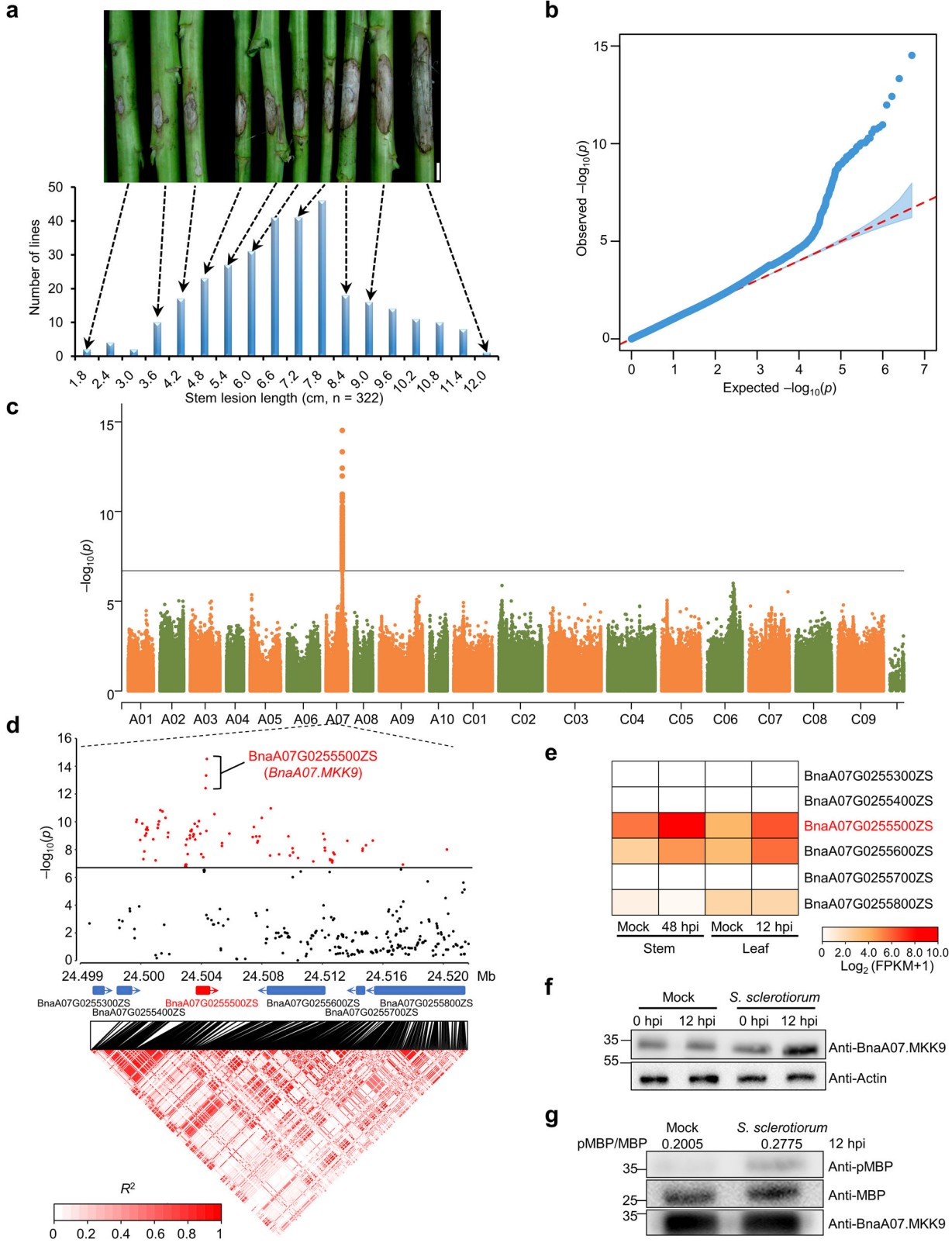

(Supplementary Data 2). New editing events were identified in the fourth homolog, *BnaCO2.MKK9*, in $T_1$ progenies of self-pollinated #5 and #7 lines (Supplementary Data 2). By $T_4$, homozygous quadruple mutants were obtained (Fig. 2e; Supplementary Fig. 3e, f; Supplementary Data 2). Further sequence analysis found no editing events in the potential off-target sites of these two *Bnamkk9* quadruple mutants (Supplementary Data 3).

When inoculated with *S. sclerotiorum*, these *Bnamkk9* loss-of-function mutants exhibited more severe symptoms than the non-mutated J9712 plants. Compared to J9712, the *Bnamkk9* mutants had significantly larger lesions on leaves (37.3–39.5%, Fig. 2f, g) and significantly longer lesions on stems (17.1–21.3%, Fig. 2h, i). Similar results were obtained in $T_5$ progenies in the subsequent year (Supplementary Fig. 3g, h), indicating that *BnaMKK9* plays a positive role in SSR

**Fig. 1 | Genome-wide association studies (GWAS) identified the gene *BnaA07.MKK9* as being associated with Sclerotinia stem rot (SSR) resistance in oilseed rape. a** Distribution of the lesion length on stems in the association panel of 322 oilseed rape accessions. Representative images of stem lesions are shown at the top. Scale bars: 2 cm. **b, c** Quantile-quantile (**b**) and Manhattan plots (**c**) illustrating the GWAS of SSR resistance, with the dashed line in (**c**) denoting the significance threshold ($-\log_{10}(P) = 6.7$). **d** Local Manhattan plot (top) and linkage disequilibrium (LD) heatmap (bottom) in the 24.499–24.520 Mb region on chromosome A07, where red marks denote SNPs of high significance. The LD scale transitions from white to red to illustrate correlation coefficient ($R^2$) values. Positions of annotated genes and the direction of their transcription are denoted by blue/red boxes with arrows. **e** Expression patterns of the candidate genes in response to *S. sclerotiorum* infection using RNA-seq data. Stem expression at 48 h post-inoculation (hpi) was acquired from our previous study[63], and leaf expression at 12 hpi was newly evaluated in this study. Expression was quantified as fragments per kilobase of transcript per million fragments mapped (FPKM). **f, g** Protein abundance (**f**) and kinase activity (**g**) of BnaA07.MKK9 following *S. sclerotiorum* inoculation compared to mock-inoculated leaves. BnaA07.MKK9 was quantified using the anti-BnaA07.MKK9 antibody, with Actin serving as a control for protein loading. Kinase function was gauged through an immunoprecipitation (IP) kinase assay using myelin basic protein (MBP) as the substrate. BnaA07.MKK9 and MBP input were confirmed via immunoblotting (IB) using the anti-BnaA07.MKK9 and anti-MBP antibodies. Phosphorylated MBP was detected with the anti-pMBP antibody. Values indicate the relative band density of phosphorylated MBP normalized to MBP protein concentration using ImageJ. Molecular mass markers in kilodaltons are shown on the left. Experiments were repeated three times with similar results. Source data are provided as a Source Data file.

resistance. Besides, the key agronomic traits of *Bnamkk9* mutants were not significantly altered in the field conditions (Supplementary Data 4). It is worth noting that two *Arabidopsis* T-DNA insertion null mutants, *Atmkk9-1* (SALK_017378) and *Atmkk9-2* (SAIL_60_H06) also had lower resistance to SSR (Supplementary Fig. 4), suggesting that the role of *MKK9* in SSR resistance is conserved across different plant species.

However, constitutive overexpression of *BnaA07.MKK9^DD^*, driven by the CaMV 35 S promoter, induced a permanent immune response that was lethal due to severe growth defects in both oilseed rape and *Arabidopsis* (Supplementary Fig. 5). Additionally, *N. benthamiana* leaves transiently expressing *BnaA07.MKK9* and *BnaA07.MKK9^DD^* exhibited cell death and accumulation of hydrogen peroxide ($H_2O_2$) at 72 and 12 h post-transfection (hpf), respectively (Fig. 2j, k). In contrast, leaves expressing the kinase-inactive allele *BnaA07.MKK9^KR^* – containing a Lys73 to Arg mutation in the ATP-binding site – and those transfected with an EV showed neither cell death nor $H_2O_2$ accumulation (Fig. 2j, k), implying a correlation between the kinase activity of BnaA07.MKK9 and hypersensitive response (HR) cell death.

At 12 hpf, RNA-seq analysis detected 147 differentially expressed genes (DEGs) in *N. benthamiana* leaves transiently expressing *BnaA07.MKK9^DD^* compared to those expressing *BnaA07.MKK9^KR^*. Gene ontology (GO) enrichment analysis revealed the DEGs' involvement in fungal defense response, immune response, and overall immune system processes (Fig. 2l). Interestingly, SSR infected *N. benthamiana* leaves transiently expressing *BnaA07.MKK9^DD^* had 34.7% smaller lesions at 24 hpi compared with those transiently expressing *BnaA07.MKK9^KR^* (Fig. 2m, n). These results suggest that activation of *BnaA07.MKK9* triggers an immune response, including $H_2O_2$ accumulation and eventual HR-related cell death. Collectively, these experiments suggest that while *BnaA07.MKK9* is a positive regulator of SSR resistance, its overactivation can lead to constitutive immunity.

## *BnaA07.MKK9* positively regulated IGS, camalexin, and ethylene production

To circumvent the negative effects of constitutive *BnaA07.MKK9* OE (as seen in Supplementary Fig. 5), we employed the methoxyfenozide (MOF)-inducible expression system to control the expression of *BnaA07.MKK9^DD^* in transgenic *Arabidopsis* plants (*Csv::BnaA07.MKK9^DD^*; Supplementary Fig. 6a). This system successfully increased *BnaA07.MKK9^DD^* expression upon applying MOF to *Csv::BnaA07.MKK9^DD^* transgenic lines (#1, #6, #11, T3 progenies), while the control Col-0 plants remained unchanged (Fig. 3a; Supplementary Fig. 6b). The induced plants also showed a significant increase in SSR resistance compared to the DMSO treatment (Fig. 3b, c), with concomitant loss of green coloration and heightened $H_2O_2$ levels 72 h after induction, indicating an immune response (Supplementary Fig. 6c, d). These findings strongly suggest that induction of *BnaA07.MKK9^DD^* expression confers resistance to SSR in *Arabidopsis*.

RNA-Seq analysis in *Csv::BnaA07.MKK9^DD^* transgenic plant #11 identified 252 up- and 22 down-regulated genes at 2 h post MOF

treatment (Fig. 3d). The number of up- and down-regulated genes increased to 2278 and 2443 at 8 h after treatment (Fig. 3e). A Kyoto Encyclopedia of Genes and Genome (KEGG) analysis revealed that these DEGs were enriched in a number of immune pathways, such as plant-pathogen interaction, glucosinolates (GSLs) biosynthesis, plant hormone signal transduction and tryptophan metabolism (Fig. 3f), hinting that *BnaA07.MKK9^DD^* triggers the biosynthesis of plant hormones and immunity-related metabolites.

To verify our speculation, we measured the levels of specific plant hormones and defense compounds in the leaves of *Csv::BnaA07.MKK9^DD^* plants. At 8 h after MOF induction, there was a significant rise in key molecules in plant defense, namely ET, camalexin, and two of the main indole GLSs (IGSs), indol-3-ylmethyl glucosinolate (I3G) and 4-methoxy-indol-3-ylmethyl glucosinolate (4MI3G), by 7.4-, 10.3-, 1.4- and 2.6-fold, respectively (Fig. 3g–i). Biosynthesis genes for ET (*AtACS2, AtACS6*), camalexin (*AtCYP71A13, AtCYP71A12, AtPAD3*), and IGS (*AtCYP81BF1, AtIGMT1* and *AtIGMT2*) were also up-regulated (Fig. 3g–i). However, there were no significant changes in the salicylic acid (SA), jasmonic acid (JA), jasmonoyl isoleucine (JA-Ile) (Supplementary Fig. 7) and aliphatic GSLs (Supplementary Fig. 8). These results reveal that activation of *BnaA07.MKK9^DD^* expression induces the biosynthesis of ET, camalexin, and IGSs.

Under control conditions, *Bnamkk9* oilseed rape plants displayed normal levels of the defense compounds camalexin, I3G, 4MI3G, 1-aminocyclopropane-1-carboxylic acid (ACC, an ET precursor), and $H_2O_2$ (Fig. 3j–n). While upon *S. sclerotiorum* challenge, the levels of these defense compounds increased significantly in J9712, they increased only slightly or not at all in the *Bnamkk9* plants (Fig. 3j–n). Collectively, these results suggest that *BnaA07.MKK9* promotes ET, camalexin, and IGS biosynthesis and $H_2O_2$ accumulation, thereby enhancing resistance to *S. sclerotiorum*.

## BnaA07.MKK9 physically interacted with and phosphorylated BnaMPK3 and BnaMPK6

The gene BnaA07.MKK9, a member of the MKK family, is known to activate downstream MPKs in response to environmental stimuli[22]. To identify which BnaMPKs are targets of BnaA07.MKK9, we conducted yeast two-hybrid (Y2H) assays. Out of the 16 cloned BnaMPKs, only weak interactions between BnaA07.MKK9^KR^ and BnaC03.MPK3/BnaC03.MPK6 were observed (Supplementary Fig. 9). However, no interactions with BnaA07.MKK9 or BnaA07.MKK9^DD^ were detected, suggesting that the active form of BnaA07.MKK9 may destabilize interactions with BnaC03.MPK3/BnaC03.MPK6 (Fig. 4a).

To confirm the interactions between BnaA07.MKK9 and BnaC03.MPK3/BnaC03.MPK6 *in planta*, we conducted luciferase complementation imaging (LCI) assays in *N. benthamiana* leaves. BnaA07.MKK9^KR^ was used for these assays to avoid the HR cell death triggered by the wild-type and DD variants of BnaA07.MKK9. The results, illustrated in Fig. 4b, showed robust luciferase activity when BnaA07.MKK9^KR^ was co-expressed with BnaC03.MPK3 or

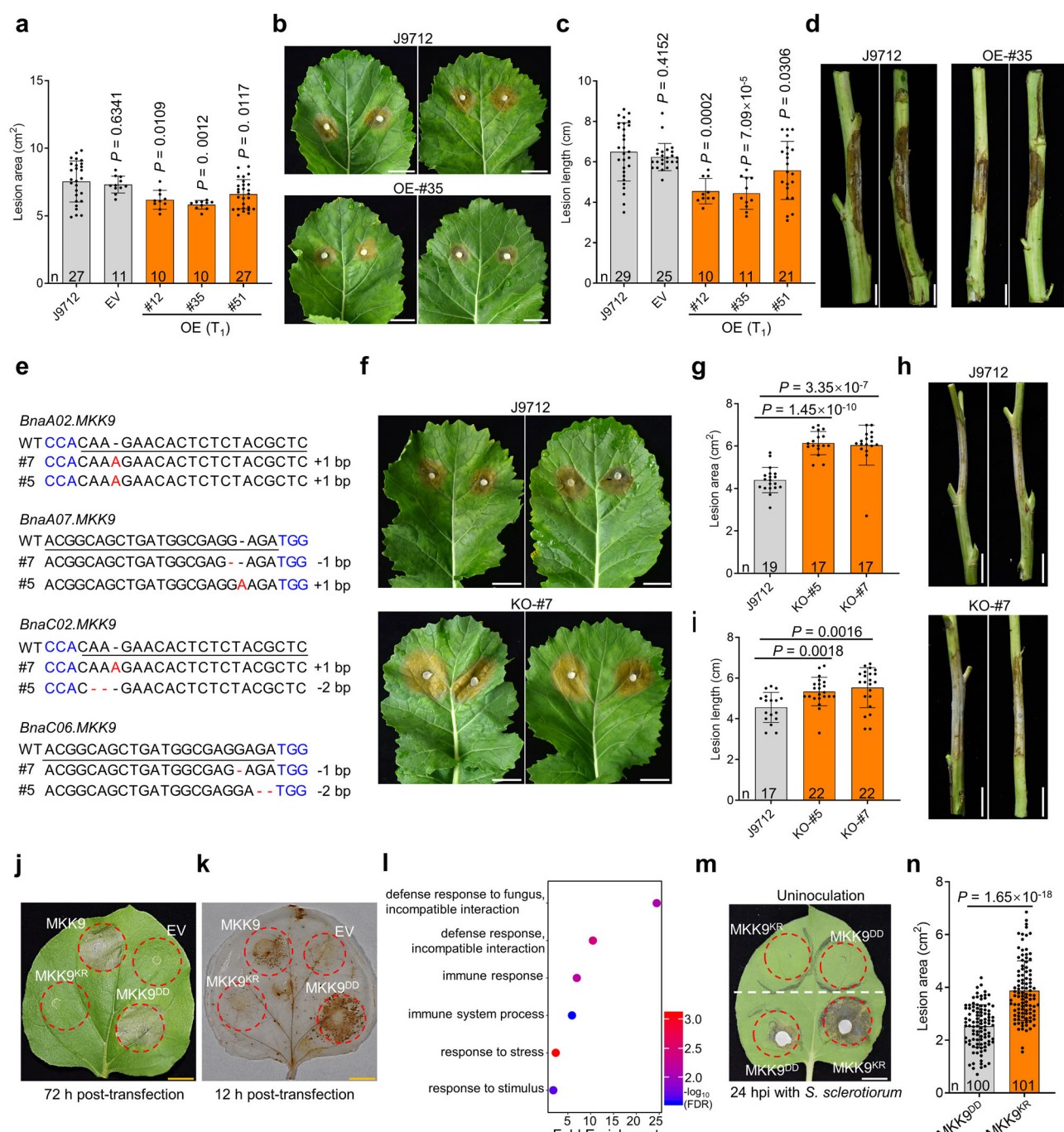

**Fig. 2 | BnaA07.MKK9 plays a positive role in the regulation of Sclerotinia stem rot (SSR) resistance in oilseed rape. a–d** Evaluation of SSR resistance in over-expression (OE) lines of *BnaA07.MKK9* (T$_1$ progeny) on seedling stage leaves (**a**, **b**) and mature stage stems (**c**, **d**). Disease lesions on leaves and stems were photographed and quantified at 48 and 168 h post-inoculation (hpi), respectively. Empty vector (EV) transgenic plants and wild-type J9712 plants were used as controls. **e** Sequence diagram of *Bnamkk9* mutants generated via CRISPR/Cas9-mediated genome editing, with target sequences underlined and the protospacer-adjacent motif (PAM) in blue. **f–i** Evaluation of SSR resistance in *Bnamkk9* mutants (T$_4$ progeny) on leaves at 48 hpi (**f**, **g**) and on stems at 168 hpi (**h**, **i**). **j**, **k** Cell death at 72 h post-transfection (hpf) (**j**) and H$_2$O$_2$ accumulation measured with 3, 3'-diamino-benzidine (DAB) staining at 12 hpf (**k**) were observed in *N. benthamiana* leaves

transiently expressing *BnaA07.MKK9*, *BnaA07.MKK9*$^{KR}$, *BnaA07.MKK9*$^{DD}$ and EV, with red circles indicating infiltration zones. **l** Gene ontology enrichment analysis of differentially expressed genes between *N. benthamiana* leaves transiently expressing the *BnaA07.MKK9*$^{DD}$ and *BnaA07.MKK9*$^{KR}$ variants at 12 hpf. **m**, **n** Evaluation of SSR resistance in *N. benthamiana* transiently expressing *BnaA07.MKK9*$^{KR}$ and *BnaA07.MKK9*$^{DD}$. *S. sclerotiorum* was inoculated in the infiltration areas at 12 hpf, and disease symptoms and lesion areas were photographed (**m**) and quantified (**n**) at 24 hpi. The upper part of the *N. benthamiana* leaf was the control without *S. sclerotiorum* inoculation. Scale bars: 1 cm (**b**, **f**, **j**, **k**, **m**) and 5 cm (**d**, **h**). In (**a**, **c**, **g**, **i**, **n**), bars represent the mean ± SD, and *n* represents the number of plants. Statistical significance was determined by a two-tailed Student's *t*-test. In (**a**, **c**, **g**, **i**), significance compared with J9712. Source data are provided as a Source Data file.

BnaC03.MPK6, indicating a strong interaction. In contrast, control pairings only exhibited baseline luciferase activity.

This interaction was further supported with bimolecular fluorescence complementation (BiFC) assays in *N. benthamiana*

leaves (Supplementary Fig. 10). Additionally, in vivo co-immunoprecipitation (Co-IP) assays with BnaA07.MKK9$^{KR}$-myc and both BnaC03.MPK3-flag and BnaC03.MPK6-flag confirmed the association, as BnaA07.MKK9$^{KR}$-myc was pulled down

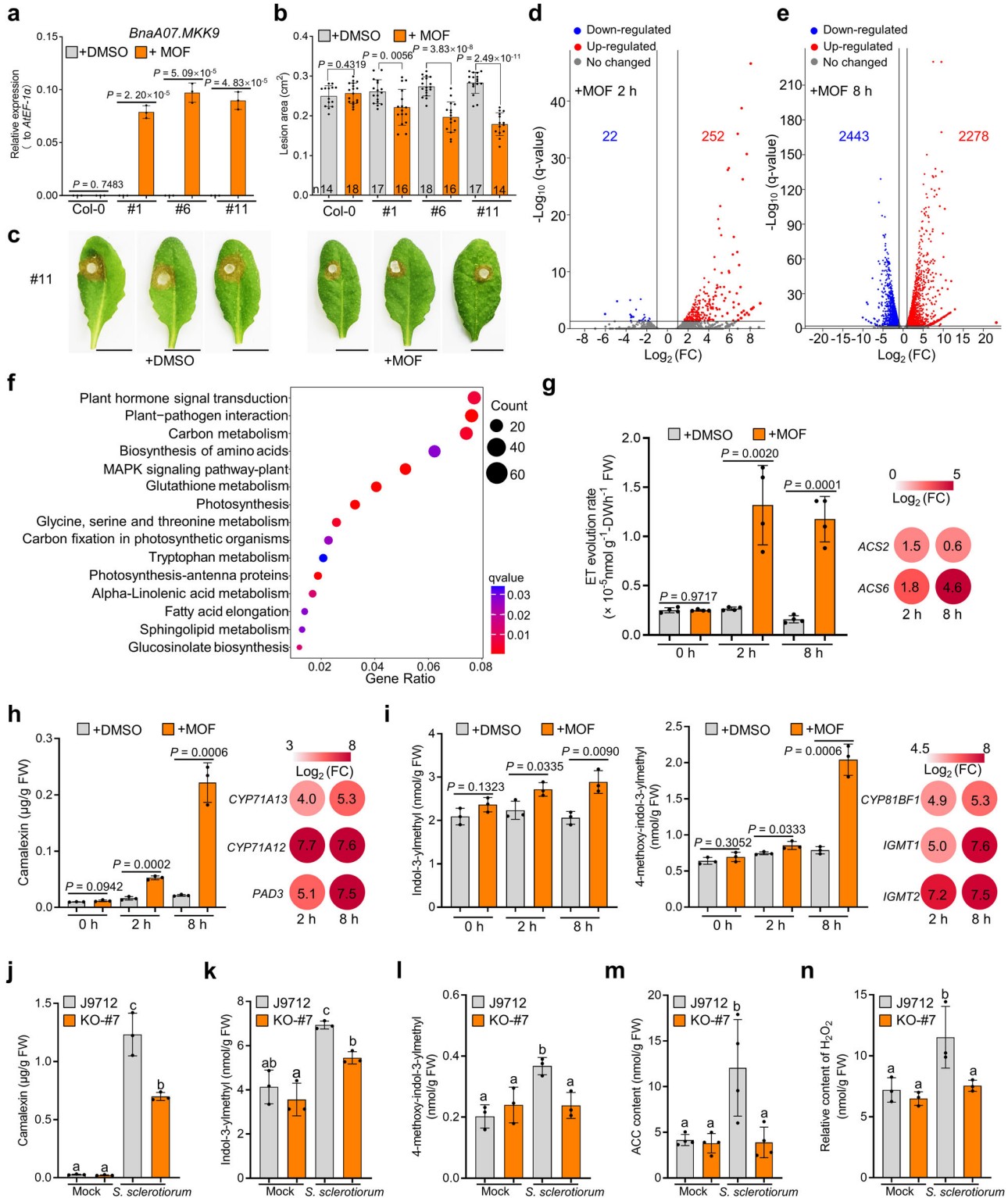

with anti-flag beads in the presence of the respective MPKs (Fig. 4c, d).

Further in vitro assays using glutathione S-transferase (GST) pull-down confirmed that BnaA07.MKK9$^{DD}$ could directly interact with both BnaC03.MPK3 and BnaC03.MPK6 (Fig. 4e, f). Collectively, these results affirm that BnaA07.MKK9 physically interacts with BnaC03.MPK3 and BnaC03.MPK6.

To determine if BnaC03.MPK3 and BnaC03.MPK6 are phosphorylated by BnaA07.MKK9, we conducted in vitro kinase assays with purified recombinant proteins. Strong phosphorylation was observed

for the kinase-inactive forms of BnaC03.MPK3 (BnaC03.MPK3$^{KR}$, with Lys$^{67}$ mutated to Arg) and BnaC03.MPK6 (BnaC03.MPK6$^{KR}$, with Lys$^{90}$ mutated to Arg) when BnaA07.MKK9$^{DD}$ was present, as depicted in Fig. 4g, h. When we altered the conventional TEY activation loop motif in BnaC03.MPK3 and BnaC03.MPK6 to AEF, no phosphorylation occurred (Fig. 4g, h), confirming that BnaA07.MKK9 targets the TEY loop for phosphorylation, as observed in past research[29].

Additionally, we analyzed the activation of BnaMPK3 and BnaMPK6 in both the *Bnamkk9* mutant and J9712 plants following *S. sclerotiorum* inoculation. The protein levels of BnaMPK3 and BnaMPK6

**Fig. 3 | Activation of *BnaA07.MKK9^DD* induces biosynthesis of ethylene (ET), camalexin, and indole glucosinolates (IGSs). a–c** Relative expression (**a**), lesion areas (**b**), and disease symptoms (**c**) of *Csv::BnaA07.MKK9^DD* transgenic *Arabidopsis* lines (#1, #6, and #11) and Col-0 treated with either DMSO (control) or 61.3 μM methoxyfenozide (MOF). These measurements were taken 8 h post-treatment, with *AtEF-1α* serving as the reference gene for expression analysis. Infected leaves were photographed and quantified at 30 h post-inoculation (hpi), scale bar: 1 cm. **d**, **e** Volcano plots showing the fold-change (FC) of differentially expressed genes (DEGs) in *Csv::BnaA07.MKK9* plants at 2 h (**d**) and 8 h (**e**) after MOF versus DMSO treatment. **f** Kyoto Encyclopedia of Genes and Genome (KEGG) pathway analysis showing the top 15 enrichment terms among the DEGs in Fig. 3e. **g–i** Quantification of ethylene (ET) (**g**), camalexin (**h**), and indol-3-ylmethyl glucosinolate (I3G) and 4-methoxyindol-3-ylmethyl glucosinolate (4MI3G) (**i**) in *Csv::BnaA07.MKK9^DD* plants at 0, 2, and 8 h after DMSO or MOF treatment. Changes in the expression of the genes involved in biosynthesis of ET, camalexin, and IGSs (MOF versus DMSO treatment) are shown as a heatmaps. **j–n** Camalexin (**j**), IGSs (**k**, **l**), 1-aminocyclopropane-1-carboxylic acid (ACC, **m**) and $H_2O_2$ (**n**) contents in *Bnamkk9* knockout (KO)-#7 and J9712 control plants at 12 hpi with *S. sclerotiorum* or the mock inoculation. For (**a**), values represent means ± SD from three biological replicates. In (**b**), bars represent the mean ± SD, and *n* represents the number of plants. In (**g–n**), bars represent the mean ± SD (*n* = 3 biological replicates in (**h–l**, and **n**), *n* = 4 biological replicates in (**g** and **m**)). In (**a**, **b**) and (**g–i**), statistical significance was determined by a two-tailed Student's *t*-test. In (**j–n**), different letters indicate significant differences (*P* < 0.05), determined with one-way analysis of variance (ANOVA) and Tukey's honestly significant difference (HSD) test. Source data are provided as a Source Data file.

remained unchanged in both the *Bnamkk9* and J9712 (Fig. 4i), however, the phosphorylated forms of BnaMPK3 and BnaMPK6, which indicate activation, were gradually elevated in J9712 from 3 hpi to 12 hpi (Fig. 4i). By contrast, this phosphorylation was noticeably reduced in *Bnamkk9* during this same time frame (Fig. 4i). Taken together, these findings imply that *S. sclerotiorum* infection triggered activation of BnaMPK6 and BnaMPK3 and that BnaA07.MKK9 increased their phosphorylation in response to *S. sclerotiorum*.

### *BnaMPK3* and *BnaMPK6* acted genetically downstream of *BnaA07.MKK9* to regulate SSR resistance

To investigate the roles of *BnaC03.MPK3* and *BnaC03.MPK6* in SSR resistance, we overexpressed *BnaC03.MPK3* and *BnaC03.MPK6* in J9712 using the CaMV 35 S promoter. The resulting OE lines ($T_0$ progeny) were then verified by qRT-PCR (Fig. 5a, d; Supplementary Fig. 11a, b). These OE lines had increased SSR resistance in both $T_1$ (Fig. 5b, c, e, f) and $T_2$ generations (Supplementary Fig. 11c, d), confirming the genes' positive impact on resistance.

To elucidate the genetic relationship between *BnaA07.MKK9* and *BnaC03.MPK3*/*BnaC03.MPK6*, we overexpressed *BnaC03.MPK3* and *BnaC03.MPK6* in the *Bnamkk9* mutant background (Fig. 5g, h; Supplementary Fig. 11e, f). Post-inoculation, the *BnaC03.MPK3* and *BnaC03.MPK6* OE lines in the *Bnamkk9* background developed disease lesions comparable to the *Bnamkk9* mutant and larger than J9712 plants (Fig. 5i, j). Similar results were observed in $T_2$ progenies (Supplementary Fig. 11g). This implies that *BnaC03.MPK3* and *BnaC03.MPK6* require *BnaMKK9* to confer SSR resistance.

The genetic relationship was also examined in *Arabidopsis*. *Atmpk3* and *Atmpk6* single mutants showed lesion areas similar to wild-type Col-0 (Supplementary Fig. 12a–d). Due to the embryo lethality of the *Atmpk3 Atmpk6* double mutant, we instead used a chemically-rescued variant, *AtMPK3SR*, which mimics the double mutant in the presence of the inhibitor NA-PP1 (Supplementary Fig. 12e). As predicted, NA-PP1 treated *AtMPK3SR* plants were more susceptible to *S. sclerotiorum* (Supplementary Fig. 12f, g), indicating the redundant roles of *AtMPK3* and *AtMPK6* in SSR resistance.

Crossing *Csv::BnaA07.MKK9^DD* with *AtMPK3SR* yielded *Csv::BnaA07.MKK9^DD*/*AtMPK3SR* plants in the F3 generation. Following treatment with MOF and NA-PP1, increased expression of *BnaA07.MKK9* was detected in these plants (Fig. 5k; Supplementary Fig. 12h). However, the enhanced expression did not translate to increased resistance; the disease response of *Csv::BnaA07.MKK9^DD*/*AtMPK3SR* was similar to the *AtMPK3SR* mutant and less effective than Col-0 (Fig. 5k–m). Combined, these findings demonstrate that *BnaMKK9* functions upstream of *BnaMPK3* and *BnaMPK6* in the SSR resistance pathway.

### Natural variation in *BnaA07.MKK9* is associated with SSR resistance in oilseed rape

To assess if sequence variations in *BnaA07.MKK9* are linked to SSR resistance in oilseed rape, we conducted haplotype analysis using the BnIR database[30]. We identified three main haplotypes, based on 4 missense and 11 synonymous SNPs (Fig. 6a), with Hap1 containing two missense mutations at the ATP-binding site. Significant differences in SSR resistance were noted among accessions with these haplotypes, with Hap0 accessions being the most resistance and those with Hap1 the least (Fig. 6b). This pattern was confirmed by testing 30 randomly selected accessions from each haplotype by stem inoculation in 2021 (Fig. 6c). Consistent with the differences observed in stem resistance between haplotypes, significant differences in leaf resistance were also observed through detached-leaf inoculations (Supplementary Fig. 13).

To explore the reasons behind these variations in resistance, we measured the kinase activities of proteins from each haplotype. All exhibited weak kinase activity after mock inoculation, but upon *S. sclerotiorum* challenge, the kinase activity of Hap0 increased significantly more than Hap2, while Hap1 showed the weakest increase (Fig. 6d; Supplementary Fig. 14). The observed kinase activity aligned with the BnaMPK3/6 phosphorylation results: phosphorylation of BnaMPK3/6 by Hap0 was much more rapid than by Hap2 and Hap1, with Hap1 being the slowest (Supplementary Fig. 15). $H_2O_2$ accumulation in *N. benthamiana* leaves transiently expressing these haplotypes further supported these differences, with Hap0 exhibiting an earlier and more robust accumulation (Fig. 6e). These results indicate that the distinct kinase activities of *BnaA07.MKK9* haplotypes affect the timing and intensity of the disease resistance response. The presence of two missense variations in the ATP-binding site of Hap1 was likely in the cause of its weaker kinase activity and diminished resistance to SSR.

Oilseed rape consists of three ecotypes: winter, semi-winter, and spring types, based on vernalization requirements. Analysis of 2311 core oilseed rape accessions from the BnIR database revealed that haplotype frequencies differ among ecotypes. Resistant Hap0 is prevalent in winter oilseed rape, while spring oilseed rape commonly carries the susceptible Hap1 (Fig. 6f). This distribution pattern aligns with observed resistance levels across ecotypes, for example, it is known that winter accessions tend to exhibit higher SSR resistance than spring accessions[31].

Geographical analysis revealed the dominance of Hap2 in Asia, Hap1 in North America, and Hap0 in Europe, Asia, and Oceania (Fig. 6g). As a result, the resistant haplotype Hap0 could be a valuable allele to enhance the SSR resistance of oilseed rape in North America, particularly in Canada. Additionally, it can also be employed to improve the SSR resistance of germplasm in Asia and Australia, as approximately 30% of these accessions lack the Hap0 haplotype.

We then investigated the haplotypes of 418 oilseed rape accessions from different breeding eras[3]. The susceptible Hap1 haplotype was present in about 10% of the accessions from the 1950s–1960s, while its prevalence rose to nearly 20% in the 1970s–1980s (Fig. 6h). This pattern coincides with a notable period in oilseed rape breeding history—the advent of double-low breeding in the 1970s. Notably,

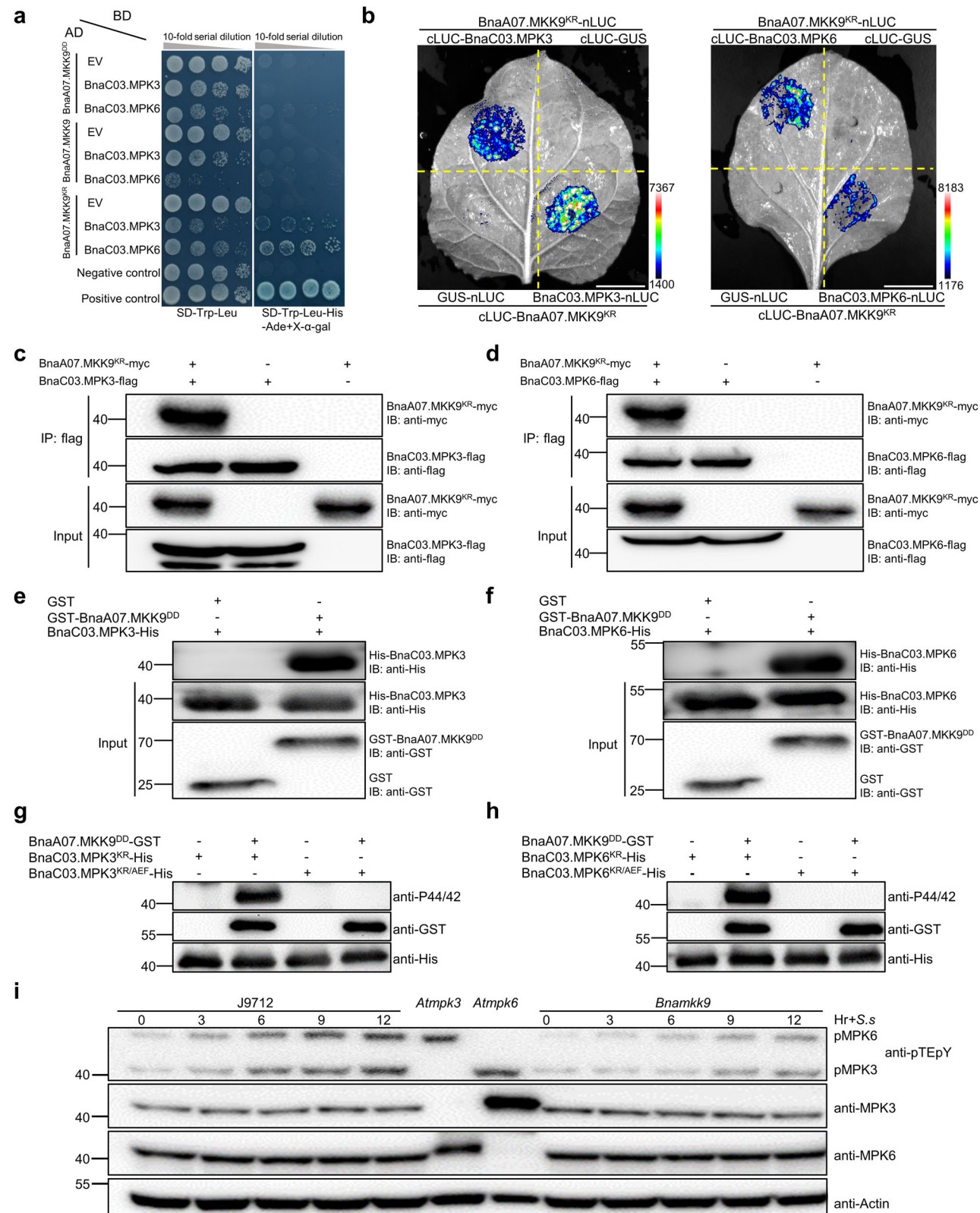

prominent 'double-low quality' donor varieties such as the erucic-acid-free 'Liho' and 'Oro,' along with the low-glucosinolate 'Bronowski,' belong to the Hap1 haplotype (https://yanglab.hzau.edu.cn/BnIR). The double-high accessions, associated with lower quality seeds, tend to be more resistant than double-low accessions, which were bred for higher quality seeds[31]. Hence, we speculate that the rise in Hap1 during the

double-low breeding era was likely an inadvertent side effect. Moving into the 1990s–2000s, with the focus shifting to breeding for high yield and enhanced resistance, the frequency of Hap1 diminished in favor of an increase in Hap0 (Fig. 6h). These trends highlight Hap0 as a beneficial haplotype for the enhancement of SSR resistance in oilseed rape, with particular relevance for double-low varieties.

**Fig. 4 | BnaA07.MKK9 interacts with and phosphorylates BnaMPK3/6 in vitro and in vivo. a** Yeast two-hybrid analysis of the interaction between BnaA07.MKK9KR and BnaC03.MPK3/BnaC03.MPK6. Yeast cells were grown on synthetic defined (SD)-Leu-Trp or SD-Leu-Trp-His-Ade/X-α-Gal medium. BD DNA-binding domain, AD activation domain. BD-53/AD-RecT was the positive control. BD-Lam/AD-RecT was the negative control. **b** Luciferase complementation imaging in *N. benthamiana* cells confirmed the interaction between BnaC03.MPK3/BnaC03.MPK6 and BnaA07.MKK9KR, with luciferase signals recorded at 40 h post-infiltration (hpf), scale bar: 1 cm. **c, d** Co-immunoprecipitation assays using anti-flag beads on protein extracts from *N. benthamiana* leaves showed the association of BnaA07.MKK9KR with BnaC03.MPK3/6. Western blotting with anti-flag and anti-myc antibodies was conducted on the input and immunoprecipitated proteins, respectively. **e, f** In vitro pull-down assays further revealed direct interactions between GST-tagged BnaA07.MKK9DD and His-tagged BnaC03.MPK3/BnaC03.MPK6, which were pulled down with anti-GST antibodies and analyzed by immunoblotting (IB) with anti-GST and anti-His antibodies, respectively. **g, h** Kinase assays conducted in vitro indicated that BnaA07.MKK9DD phosphorylates BnaC03.MPK3/BnaC03.MPK6, with phosphorylation detected by IB using the anti-pTEpY antibody and the input proteins detected with anti-GST and anti-His antibodies. **i** In *Bnamkk9* mutants, phosphorylation of BnaMPK3/6 was reduced compared to J9712 controls after *S. sclerotiorum* inoculation, with phosphorylation detected by IB with the anti-pTEpY antibody and protein levels detected with anti-BnaMPK6 and anti-BnaMPK3 antibodies. Actin served as the loading control, with *Atmpk3* and *Atmpk6* mutants providing a reference for the phosphorylation bands of MPK6 and MPK3. Hr+*S.s* represents the time, in hours, at which inoculation with *S. sclerotiorum* was conducted. In (**c–i**), the molecular mass markers in kilodaltons are shown on the left. All experiments were repeated three times, yielding consistent results. Source data are provided as a Source Data file.

## Discussion

Resistance to biotrophs, hemibiotrophs, and host-specific necrotrophs in plants often relies on simple inherited traits, whereas combatting broad host-range necrotrophs (BHNs) like *S. sclerotiorum* – an archetypical BHN affecting over 700 plant species, including key oil-producing crops like oilseed rape, soybean, and sunflower[32] – requires a more complex response involving multiple genes[19,33]. Despite extensive research, including linkage analyses and GWAS that identified numerous QTLs associated with *S. sclerotiorum*-resistant oil-producing crops[9,34,35], the genetic basis of resistance is not fully understood. This is mainly because no single resistance gene has been successfully isolated through forward genetics, primarily due to the subtle effects of each QTL.

Our study took a comprehensive approach, integrating GWAS, transcriptomics, and reverse genetics, to identify a key regulator of SSR resistance in oilseed rape, *BnaA07.MKK9*. Natural variation in this gene constitutes a valuable genetic resource for breeding SSR resistance in oilseed rape. Additionally, our results reveal a previously undiscovered role of MKK9-MPK3/6 cascade in plant defense response.

Previous research in *Arabidopsis* has shown that *MKK9* is involved in plant responses to various stimuli, such as salt stress[26], leaf senescence[27], and phosphate acquisition[28]. In this study, genetic and molecular evidence demonstrated that MKK9 plays a crucial role in SSR resistance in oilseed rape. BnaA07.MKK9 acted upstream of BnaC03.MPK3/6, by phosphorylating and thereby activating these proteins, which in turn positively impacted SSR resistance (Figs. 4, 5). These findings are consistent with earlier studies showing the positive role of *BnaMPK3/6* in SSR resistance[36–38]. While previous studies identified MKK4/5 as the upstream MKKs of MPK3/6 in rice, *Arabidopsis*, wheat, and oilseed rape[38–40], our findings add MKK9 to this list. The role of MKK9 explains why phosphorylation of MPK3/6 was reduced, but not completely absent in *mkk4/5* and *mkk9* mutants[40].

MAPK activation is an early response in plants exposed to a pathogen, playing a pivotal role in plant defense[41]. Two *Arabidopsis* MAPK cascades, MEKK1–MKK1/2–MPK4/MPK11 and MKKK3/5–MKK4/5–MPK3/6 are known to contribute to plant immunity[22]. Our study uncovered the significant role of the MKK9-MPK3/6 cascade in defending *Arabidopsis* and oilseed rape against the necrotrophic pathogen *S. sclerotiorum*. We found that when these plants encounter *S. sclerotiorum*, BnaA07.MKK9 becomes activated and subsequently phosphorylates BnaC03.MPK3/6, which in turn triggers biosynthesis of ET, camalexin, and IGSs, and promotes $H_2O_2$ accumulation and the HR. These reactions collectively enhance resistance against *S. sclerotiorum* (Fig. 7). Our findings provide a more comprehensive understanding of MAPK cascades in plant immune signaling, especially regarding resistance to necrotrophic pathogens.

Cloned genes and beneficial natural alleles are important resources for targeted trait improvement in plants[42]. We found that the Hap0 allele of *BnaA07.MKK9* is particularly effective in providing resistance against the pathogen *S. sclerotiorum*. Accessions carrying Hap0 exhibited a significant reduction in the length of disease lesions on their stems – by 33.3%–42.1% compared with Hap1, and 22.8%–28.4% compared with Hap2 (Fig. 6b, c). This enhanced resistance was attributed to the higher kinase activity of Hap0, which provided a quicker and more robust defense response against the pathogen (Figs. 6d, e, 7).

Among the three oilseed rape ecotypes, our research reveals that most winter accessions harbor the Hap0 haplotype, while approximately 30% of semi-winter and 72% of spring-type accessions lack this advantageous haplotype (Fig. 6f). This difference in frequency of the resistant Hap0 haplotype may, in part, explain previous observations that winter accessions are often more resistant to SSR than spring accessions[31].

Evidence also suggests that the double-low breeding process might have inadvertently increased the proportion of the less resistant Hap1 haplotype in some varieties (Fig. 6h). Fortunately, the genes governing the desirable 'double-low' quality (low erucic acid and glucosinolates) in oilseed rape[2,3] are not genetically linked to *BnaA07.MKK9*. This means that breeding for SSR resistance using the Hap0 haplotype of *BnaA07.MKK9* should not negatively affect the double-low qualities of the crop. Therefore, the Hap0 haplotype of *BnaA07.MKK9* could potentially serve as a promising target for improving SSR resistance, especially in spring-type, semi-winter-type, and certain double-low-quality oilseed rape varieties. Our findings additionally suggest that Hap0 had been subjected to positive selection in modern breeding programs.

In summary, *BnaA07.MKK9* plays a pivotal role in conferring resistance to *S. sclerotiorum* in oilseed rape. We have also developed a model explaining how different natural alleles of *BnaA07.MKK9* result in varying degrees of resistance to SSR in oilseed rape (Fig. 7). These findings provide valuable insights into the genetic and molecular mechanisms underlying SSR resistance, thereby holding the potential to facilitate the engineering of novel diversity for future breeding in oilseed rape.

## Methods
### Plant materials and growth conditions
The association panel used in this study consisted of 322 oilseed rape accessions, with a distribution of 286 accessions from Asia, 28 from Europe, 4 from North America, 3 from Oceania, and 1 from South America (Supplementary Data 1). These accessions were collected and supplied by the National Key Laboratory of Crop Genetic Improvement, Huazhong Agricultural University, China. Accessions with a limited range of flowering times (days from sowing to flowering, Supplementary Data 1) were selected to minimize the potential impact of developmental variations due to differences in flowering time. The semi-winter-type oilseed rape pure line J9712 was used as the transformation receptor and belongs to Hap0. Both the association panel and all transgenic

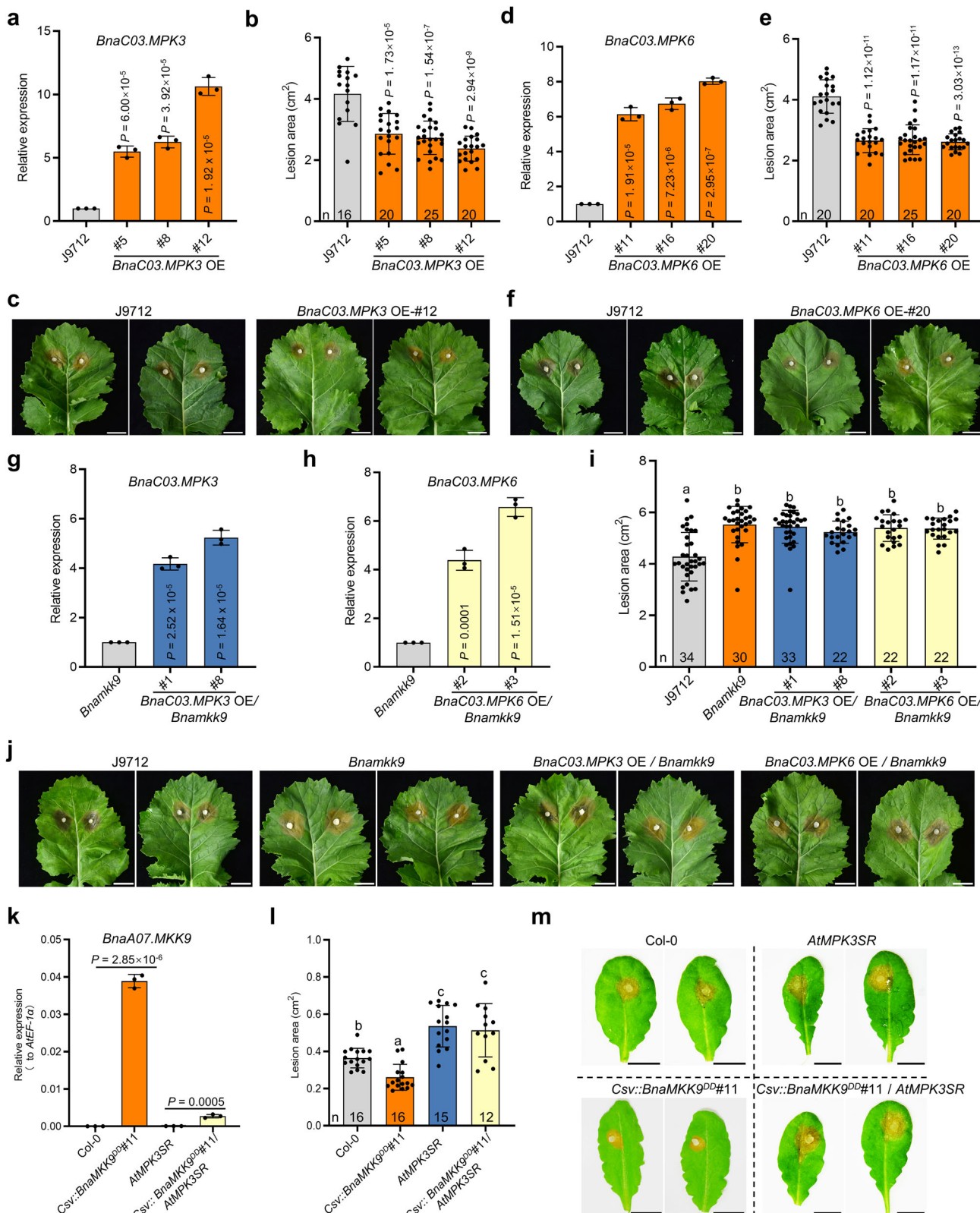

plants were cultivated in an experimental field at Yangzhou University, Jiangsu, China. Field experiments were carried out with a randomized complete block design, featuring two replications for the association panel and three for the transgenic plants.

*A. thaliana* mutants and transgenic plants used in this study were all in the Columbia-0 (Col-0) background. Mutants used in this study, *Atmkk9* (SAIL_060_H06), *Atmpk3* (SALK_151594), and

*AtMPK3SR* (genotype: *mpk3 mpk6 P_MPK3:MPK3^TG*), were described by Xu et al.[26,43]. The *Atmkk9* (SALK_017378) and *Atmpk6* (SALK_127507) mutants were obtained from the Arabidopsis Biological Resource Center (ABRC, http://www.arabidopsis.org/abrc/) and confirmed with T-DNA border primers and gene-specific primers. *A. thaliana* seeds were stratified at 4 °C in the dark for 2 days, sown on nutrient soil, and grown in a growth chamber under a 16 h

**Fig. 5 | BnaC03.MPK3 and BnaC03.MPK6 act genetically downstream of BnaA07.MKK9 to regulate Sclerotinia stem rot (SSR) resistance. a–f** Relative expression (**a**, **d**), lesion areas (**b**, **e**), and disease symptoms (**c**, **f**) of BnaC03.MPK3/BnaC03.MPK6 overexpression (OE) transgenic plants and J9712. **g–j** Relative expression (**g**, **h**), lesion areas (**i**), and disease symptoms (**j**) of J9712, Bnamkk9, BnaC03.MPK3 OE/Bnamkk9, and BnaC03.MPK6 OE/Bnamkk9. For qRT-PCR in (**a**, **d**, **g**, **h**), BnaUBC9 was the reference gene. **k–m** Relative expression (**k**), lesion areas (**l**), and disease symptoms (**m**) of Col-0, Csv::BnaA07.MKK9$^{DD}$, AtMPK3SR and Csv::BnaA07.MKK9$^{DD}$/AtMPK3SR Arabidopsis plants. qRT-PCR was performed at 12 h after methoxyfenozide (MOF) and NA-PP1 treatment. AtEF-1a served as the reference gene. The S. sclerotiorum inoculated leaves were photographed and quantified at 30 h post-inoculation (hpi). Scale bar: 1 cm. In (**b**, **c**, **e**, **f**, **i**, **j**), the S. sclerotiorum inoculated leaves were photographed and quantified at 48 hpi. Scale bar: 1 cm. In (**a**, **d**, **g**, **h**, **k**), values are means ± SD of three biological replicates. In (**b**, **e**, **i**, **l**), each bar represents the mean ± SD, and n represents the number of plants. In (**a**, **b**, **d**, **e**, **g**, **h**, **k**), statistical significance was determined by a two-tailed Student's t-test. In (**a**, **b**, **d**, **e**,), significance compared with J9712. In (**g**, **h**), significance compared with Bnamkk9.In (**i**, **l**), different lower-case letters indicate significant differences ($P < 0.05$), determined with one-way analysis of variance (ANOVA) and Tukey's honestly significant difference (HSD) post hoc test. Source data are provided as a Source Data file.

light/8 h dark photoperiod (~300 μmol·m$^{-2}$·s$^{-1}$) at 22 °C during the day and 20 °C at night, at 60% relative humidity.

## Population structure and genome-wide association analyses

The association panel SNPs were obtained through whole genome re-sequencing and comparison with genomes in the BnIR database[30]. SNPs with >30% missing data and minor allele frequencies (<0.05) were excluded.

To minimize the contribution from regions of extensive strong LD, we scanned the whole genome with a sliding window of 50 kb (in steps of 10 SNPs) and removed any SNPs correlated with other SNPs within the window with a correlation coefficient ($R^2$) > 0.2 using the PLINK software (v.1.90b4.4)[44]. Subsequently, we constructed a phylogenetic tree using FastTree[45] with default parameters. The population's genetic structure was assessed using ADMIXTURE (v.1.3.0) software[46] with K values ranging from 2 to 7. Ultimately, K = 2 was chosen for subsequent analysis. Principal component analysis (PCA) was carried out using the GCTA software (v.1.94.0)[47].

GWAS analysis was conducted using a mixed linear model with GEMMA (v0.98.1)[48]. The cutoff for determining significant associations was set to −log10(1/n), where 'n' represents the total number of SNPs. To visualize GWAS results, Manhattan plots and QQ plots were generated using the CMplot package (v.4.0) in R (v.x64 4.1.1) (https://github.com/YinLiLin/R-CMplot/blob/master/CMplot.r). Linkage disequilibrium (LD) between pairs of SNPs in the regions surrounding the three most significant SNPs (leading SNPs) was estimated by the correlation coefficient ($R^2$) using PLINK (v.1.90b4.4)[44] and visualized with LDheatmap[49].

## Haplotype and phylogenetic analysis

Haplotype analysis of BnaA07.MKK9 in 2311 core oilseed rape accessions were performed via the BnIR database[30]. For phylogenetic analysis, the predicted amino acid sequences were aligned using ClustalW (https://www.genome.jp/tools-bin/clustalw). A phylogenetic tree was constructed by the neighbor-joining method with 1000 bootstrap replicates in the MEGA software (v.7.0.21)[50].

## Plasmid construction and plant transformation

The full-length coding sequences (CDSs) of BnaA07.MKK9 (Hap0), BnaC03.MPK3 and BnaC03.MPK6 were amplified from J9712. The full-length coding sequences (CDSs) of BnaA07.MKK9$^{Hap1}$ and BnaA07.MKK9$^{Hap2}$ were amplified from Niklas and SWU47, respectively. BnaA07.MKK9$^{DD}$, BnaA07.MKK9$^{KR}$, BnaC03.MPK3$^{KR}$, BnaC03.MPK3$^{KR/AEF}$, BnaC03.MPK6$^{KR}$ and BnaC03.MPK6$^{KR/AEF}$ variants were generated by site-directed mutagenesis using the overlap extension PCR technique[51].

To construct the OE vectors, the CDSs of BnaA07.MKK9$^{DD}$, BnaC03.MPK3 and BnaC03.MPK6 were individually inserted into the PMDC83 vector under the control of the CaMV 35S promoter. To generate the knockout vector, four specific sgRNAs targeting BnaMKK9 were designed by CRISPR-P v2.0 (http://crispr.hzau.edu.cn/cgi-bin/CRISPR2/). sgRNA arrays were cloned into a pYLCRISPR/Cas9 multiplex genome targeting vector according to Ma et al.[52].

All vectors were transformed into A. tumefaciens strain GV3101 and further transformed into oilseed rape line J9712 with an A. tumefaciens-mediated hypocotyl method[53]. Positive transgenic plants were identified by PCR. The Bnamkk9 knockout (KO) mutants were identified by PCR and Sanger sequencing of the target regions. For off-target analysis, the potential off-target sites were searched by CRISPR-P v2.0. The sequences at these sites were amplified by gene-specific primers and validated by Sanger sequencing.

The transgene-free Bnamkk9 mutant was isolated and used to generate BnaC03.MPK3 OE/Bnamkk9 and BnaC03.MPK6 OE/Bnamkk9 plants.

The MOF-inducible expression system was used to selectively express BnaA07.MKK9$^{DD}$ in Arabidopsis[54]. Transgenic Arabidopsis plants, Csv::BnaA07.MKK9$^{DD}$ were generated by A. tumefaciens-mediated floral dipping in the Col-0 background[55] and Csv::BnaA07.MKK9$^{DD}$/AtMPK3SR plants were generated by genetic crossing.

The above vectors, except for Csv::BnaA07.MKK9$^{DD}$ were generated via homologous recombination using a ClonExpress II One Step Cloning Kit (Vazyme, China). For Csv::BnaA07.MKK9$^{DD}$, an entry vector was constructed by cloning the CDS of BnaA07.MKK9$^{DD}$ into the pENTR/D-TOPO® vector (Invitrogen, Carlsbad, CA, USA), following the manufacturer's protocol. Then, via LR recombination, the pENTR-BnaA07.MKK9$^{DD}$ was transferred into the homologous recombination site of the MOF-inducible expression vector[54] to generate Csv::BnaA07.MKK9$^{DD}$, according to the manufacturer's protocol (Invitrogen, USA).

## Assessment of S. sclerotiorum resistance

The S. sclerotiorum isolate SS-1 was cultured on potato dextrose agar (PDA, Becton, Dickinson and Company, USA) at 23 °C in the dark. Inoculum was prepared by punching a mycelial agar plug with a diameter of 7 mm from the actively growing margin of a 2-day-old S. sclerotiorum culture. Mock-inoculated leaves were treated with 7 mm in diameter agar plugs. For the association panel, the stems of 6−8 individuals of each accession were inoculated in each replicate at the mature plant stage in the field. The stems were inoculated at a height of 50 cm above the ground one week after the termination of flowering. For the transgenic oilseed rape plants, alongside stem inoculation, we inoculated the last or penultimate fully extended leaves of six- to eight-week-old field-grown plants in a growth room. The procedures for detached-leaf inoculation were previously described[56]. The mycelial agar plug was inoculated on the middle of each leaf. The inoculated leaves were placed in the plastic tray and covered with plastic film. The inoculated leaves were kept at 22 °C in dark. The lesion length on stems and the lesion area on leaves were measured at 168 and 48 hpi, respectively.

For S. sclerotiorum inoculation in Arabidopsis, 4-week-old unfolded leaves were detached and placed on 2 mm in diameter mycelial agar plugs for leaf inoculation following the method outlined by Lin et al.[57]. The inoculated leaves were photographed and quantified at 30 hpi.

## Y2H assay

The Y2H assay was performed according to the Yeast Protocol Handbook (Clontech). Briefly, the CDSs of BnaA07.MKK9,

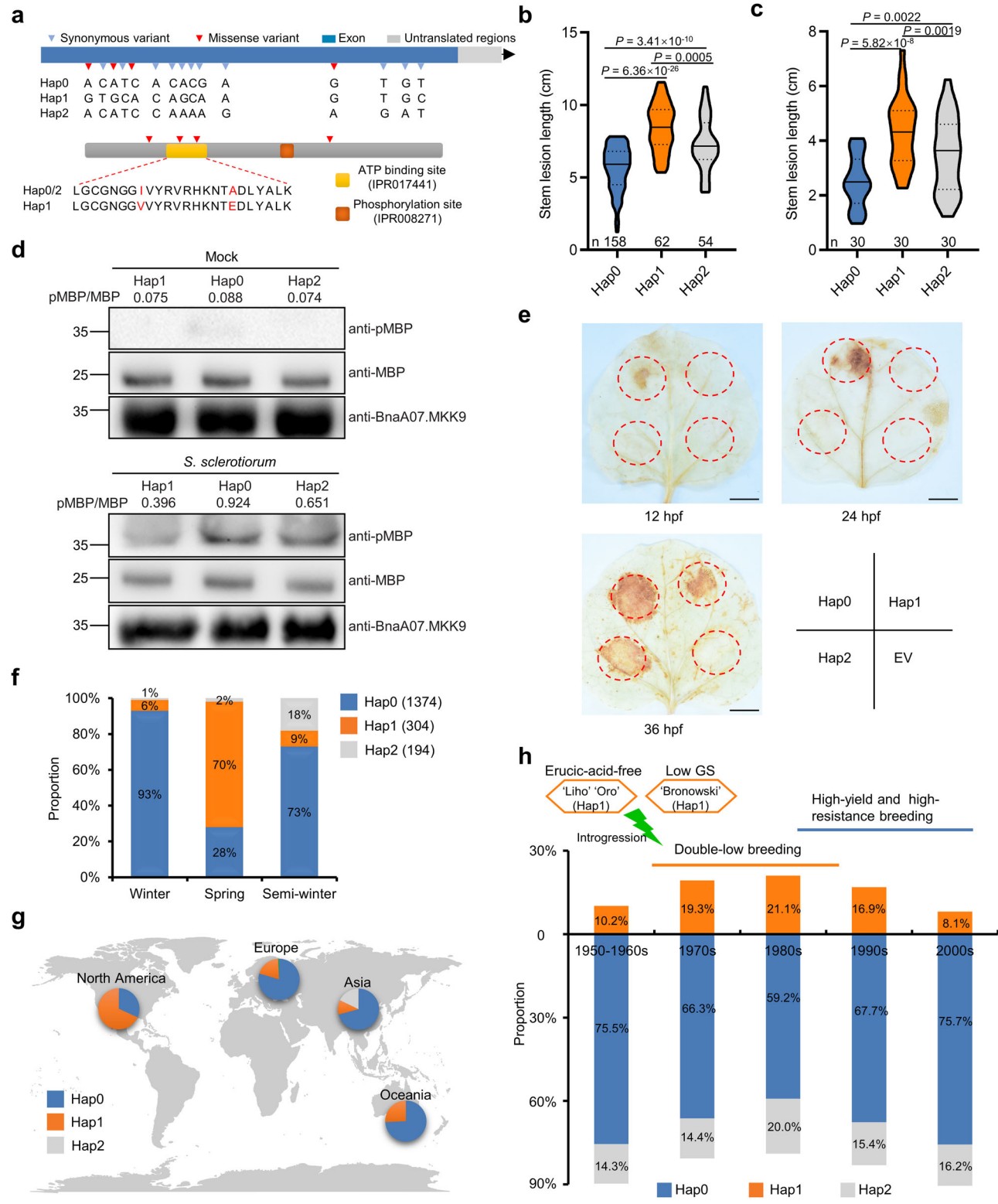

*BnaA07.MKK9^DD* and *BnaA07.MKK9^KR* were individually cloned into the pGADT7 vector, while the CDS of *BnaC03.MPK3* and *BnaC03.MPK6* were individually cloned into the pGBKT7 vector. BD (DNA-binding domain) and AD (activation domain) fusion constructs were co-transformed into the yeast strain AH109 and cultured at 30 °C for 3 d on the synthetic defined (SD)/-Trp-Leu medium. Dilutions of the yeast cultures ($10^{-0}$, $10^{-1}$, $10^{-2}$, and $10^{-3}$) were spotted on SD/-Trp-Leu-His-Ade medium containing X-α-gal (40 mg/L) for protein interaction tests.

## BiFC assays

BiFC assays were performed as previously described[58]. The CDSs of *BnaA07.MKK9^KR* (without a stop codon) and *BnaC03.MPK3*/*BnaC03.MPK6* were cloned into the pSPYCE and pSPYNE vectors, respectively. Combinations of vectors were transiently co-expressed in 4-week-old *N. benthamiana* leaves via *A. tumefaciens*-mediated infiltration[59]. YFP fluorescence was visualized at 2 days after infiltration using a confocal laser scanning microscope (ZEISS LSM 880NL0, Oberkochen, Germany).

**Fig. 6 | Natural variants of *BnaA07.MKK9* are associated with Sclerotinia stem rot (SSR) resistance in oilseed rape. a** Haplotype analysis of *BnaA07.MKK9* based on the BnIR database (https://yanglab.hzau.edu.cn/BnIR). Top: gene structure and DNA polymorphism of *BnaA07.MKK9*, with exons shown as blue boxes. Bottom: schematic diagram of amino acid motifs. ATP-binding site and phosphorylation sites were predicted with the InterPro database (https://www.ebi.ac.uk/interpro/). **b** SSR resistance of three haplotypes among 322 oilseed rape accessions in 2016. **c** SSR resistance of 30 randomly selected accessions for each haplotype in 2021. The violin plots in (**b**, **c**) show the median (the solid line in the middle) and the lower and upper quartiles (the dashed lines). The shape of the violin plots represents data distribution, and the bounds indicate the minima and maxima. Statistical significance was determined by a two-tailed Student's *t*-test. **d** Kinase activity of BnaA07.MKK9 in each haplotype in *S. sclerotiorum* and mock-inoculated leaves analyzed by the immunoprecipitation (IP) kinase assay using myelin basic protein

(MBP) as substrate. Values indicate the relative band density of phosphorylated MBP normalized to MBP protein concentration using ImageJ. Molecular mass markers in kilodaltons are shown on the left. For each haplotype, samples were pooled from 10 randomly selected oilseed rape accessions. **e** Differences in $H_2O_2$ accumulation induced by the three haplotypes, after their transient expression in *N. benthamiana* leaves at 12, 24, and 36 h post-transfection (hpf), respectively. Scale bar: 1 cm; EV empty vector. **f** Variation in haplotype frequency among different ecotypes. **g** Geographic distribution of the three haplotypes. The global map was generated using the map data function in ggplot2 packages in R (v 4.3.2). For (**f**, **g**) 2311 core oilseed rape accessions from the BnIR database were analyzed. **h** The frequency of haplotypes in 418 accessions from different breeding periods. The green lightning symbol represents introgression from erucic-acid-free and low-glucosinolate (GS) varieties. In (**d**, **e**), experiments were repeated three times with similar results. Source data are provided as a Source Data file.

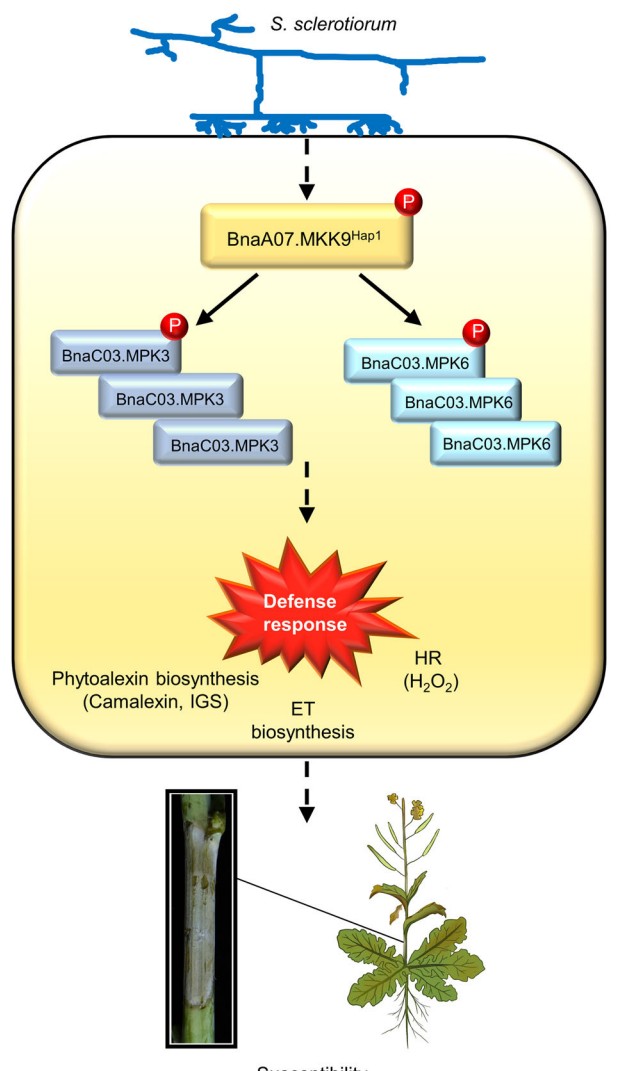

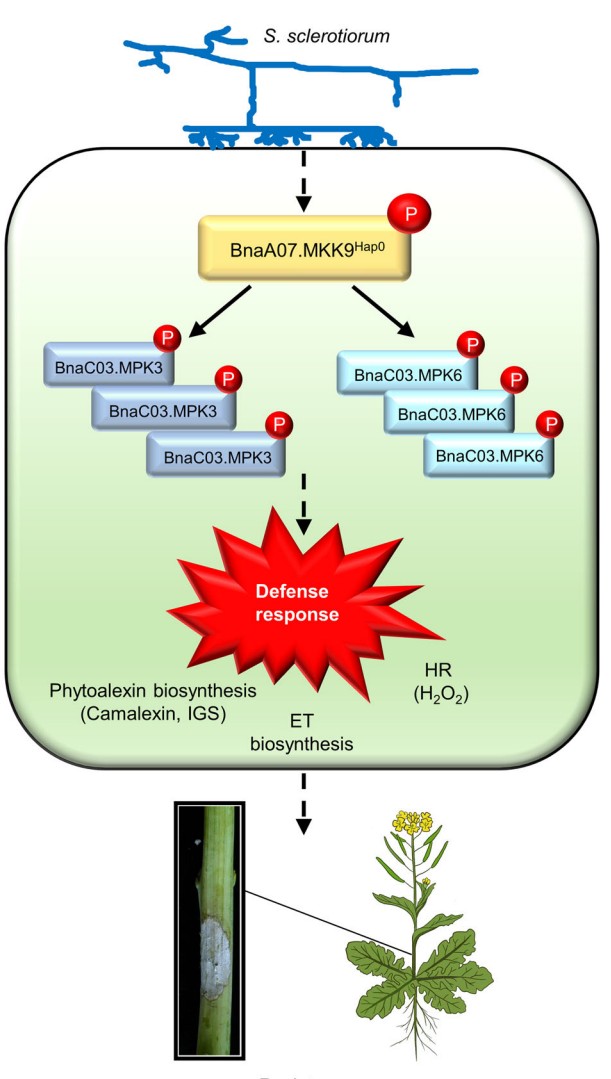

**Fig. 7 | Proposed model for the role of *BnaA07.MKK9* in Sclerotinia stem rot (SSR) resistance.** Upon S. sclerotiorum challenge, the oilseed rape accessions carrying Bna*A07.MKK9^Hap0^* showed higher kinase activity and enhanced BnaC03.MPK3/6 phosphorylation, while the oilseed rape accessions carrying Bna*A07.MKK9^Hap1^* showed weaker kinase activity and decreased BnaC03.MPK3/6

phosphorylation. As a result, compared with Hap1, Hap0 accessions had earlier and more robust defense responses that initiated biosynthesis of ethylene (ET), camalexin, and indole glucosinolates (IGSs) and promoted accumulation of $H_2O_2$ and the hypersensitive response (HR), ultimately resulting in enhanced SSR resistance. The circle with the letter 'P' within it represents phosphorylation.

## LCI assays

LCI assays were performed as described previously[60]. The CDSs of *BnaA07.MKK9^KR^* (without a stop codon) and *BnaC03.MPK3/BnaCO3.MPK6* were cloned into the nLUC and cLUC vectors, respectively. *N. benthamiana* leaves were co-

transformed with the nLUC and cLUC fusion constructs via *A. tumefaciens*-mediated infiltration[59] and then cultured for 2 d. The luciferase signal was captured using a Tanon 5200 image system (Tanon Science & Technology, Shanghai, China) after spraying with 1 mM D-luciferin (Sigma–Aldrich, St. Louis, MO, USA).

## Protein expression and purification from *Escherichia coli*

To generate *GST-BnaA07.MKK9$^{DD}$*, *GST-BnaA07.MKK9$^{Hap0}$*, *GST-BnaA07.MKK9$^{Hap1}$*, and *GST-BnaA07.MKK9$^{Hap2}$* constructs, the CDS of each gene was cloned into a pGEX6P-1 vector. Similarly, to generate *BnaC03.MPK3-His*, *BnaC03.MPK3$^{KR}$-His*, *BnaC03.MPK3$^{KR/AEF}$-His*, *BnaC03.MPK6-His*, *BnaC03.MPK6$^{KR}$-His*, and *BnaC03.MPK3$^{KR/AEF}$-His* constructs, the CDS of each gene was cloned into a pET30-a vector.

The resulting plasmids were transformed into *E. coli* BL21 (DE3) cells and recombinant protein expression was induced with 0.5 mM β-D-1-thiogalactopyranoside (IPTG) at 16 °C overnight. Recombinant proteins were purified by using the MagnetGST™ and MagnetHis™ protein purification systems (Promega, Madison, WI, USA) according to the manufacturers' instructions.

## IB analysis and antibody preparation

For IB analysis, total proteins were separated by 10% (w/v) SDS-polyacrylamide gel electrophoresis (PAGE) and were transferred to a polyvinylidene difluoride (PVDF) membrane (Millipore, Bedford, MA, USA). The membrane was blocked with PBST buffer (140 mM NaCl, 10 mM KCl, 2 mM KH$_2$PO$_4$, 8 mM Na$_2$HPO$_4$, 0.1% [v/v] Tween 20) containing 5% (w/v) nonfat dry milk and incubated for 2 h at room temperature with gentle shaking. The membrane was then incubated with specific primary antibodies at 4 °C overnight. The membrane was subsequently washed with PBST buffer and incubated with horseradish peroxidase (HRP)-conjugated secondary antibodies. Chemiluminescent signals of protein bands were detected with SuperSignal™ West Pico PLUS chemiluminescent substrate (Thermo Fisher Scientific, Waltham, MA, USA). Images were captured with a Tanon 5200 chemiluminescent imaging system (Tanon, China).

Primary antibodies used in this study include the anti-GST antibody (Cowin Bio, Jiangsu, China, CW0084), anti-His antibody (Cowin Bio, CW0285), anti-Myc antibody (MBL, Nagoya, Japan, M192-3S), anti-DDDDK tag antibody (MBL, Japan, M185-3L), anti-β-Actin antibody (TransGen Biotech, Beijing, China, HC201-01), anti-pTEpY antibody (Cell Signaling Technology, Beverly, MA, USA, 9101S), anti-pMBP antibody (Millipore, Eschborn, Germany, 05-429) and the anti-MBP antibody (Sigma–Aldrich, M3821). The anti-BnaMPK3, anti-BnaMPK6 and anti-BnaA07.MKK9 antibodies used in this study were provided by Kaijing Biotech (Shanghai, China). The anti-BnaMPK3 antibody was raised against a peptide from BnaMPK3 (TESDLGFTHNEDAKR). The anti-BnaMPK6 antibody was raised against TPSEQEVEFLNENAK from BnaMPK6. The anti-BnaA07.MKK9 antibody was raised against the BnaA07.MKK9 protein. Secondary antibodies included HRP-conjugated goat anti-rabbit IgG (Cowin Bio, CW0103S) and HRP-conjugated goat anti-mouse IgG (Cowin Bio, CW0102S). Primary and secondary antibodies were used at dilutions of 1:1,000 and 1:5,000, respectively.

## GST-pull-down assays

For pull-down assays, GST or GST-BnaA07.MKK9$^{DD}$ proteins immobilized on magnetGST particles (Promega) were incubated with BnaC03.MPK3-His or BnaC03.MPK6-His proteins in buffer (2 mM KH$_2$PO$_4$, 4.2 mM Na$_2$HPO$_4$, 10 mM KCl, 140 mM NaCl, pH 7.5) at 4 °C for 2 h. The particles were washed three times with PBS (containing 0.1% Triton X-100 [v/v]), separated on 10% SDS-PAGE gels, and detected by IB with anti-His and anti-GST antibodies.

## Co-IP assays

For the Co-IP assays, the CDSs of *BnaC03.MPK3* and *BnaC03.MPK6* were cloned into p1300-221-flag vectors. The CDS of *BnaA07.MKK9$^{KR}$* without its stop codon was cloned into the p1300-221-myc vector. The resulting constructs were transformed into *A. tumefaciens* stain GV3101. The *BnaC03.MPK3*-flag and *BnaA07.MKK9$^{KR}$*-myc vectors or *BnaC03.MPK6*-flag and *BnaA07.MKK9$^{KR}$*-myc vectors were transiently co-transformed into *N. benthamiana* leaves via *A. tumefaciens*-mediated infiltration[59]. Thirty-six h after infiltration, the total proteins were extracted with buffer (150 mM NaCl, 50 mM Tris-HCl [pH 7.5], 5 mM dithiothreitol [DTT], 1 mM EDTA, 0.3% IGEPAL® CA-630 (Sigma–Aldrich), and 1× protease inhibitor cocktail) and centrifuged at 13,500 × *g* for 30 min at 4 °C. The supernatants were incubated with anti-flag magnetic beads (Beyotime Biotech, Shanghai, China) at 4 °C overnight, and then the beads were collected and washed three times with IP buffer. Immunocomplexes retained on the beads were detected by IB with the anti-Myc antibody.

## In vitro phosphorylation assays

For the in vitro phosphorylation assays, 20 μg BnaC03.MPK3$^{KR}$-His, BnaC03.MPK6$^{KR}$-His, BnaC03.MPK3$^{KR/AEF}$-His or BnaC03.MPK6$^{KR/AEF}$-His was incubated with 5 μg GST-BnaA07.MKK9$^{DD}$ at 30 °C for 30 min in kinase reaction buffer (25 mM Tris-HCl [pH 7.5], 10 mM MgCl$_2$, 1 mM DTT, 50 μM ATP). Phosphorylation of BnaC03.MPK3 and BnaC03.MPK6 was detected by IB with the anti-pTEpY antibody. Loading controls were detected with the anti-His and anti-GST antibodies.

For the in vitro phosphorylation assay of BnaA07.MKK9 corresponding to each haplotype, the CDSs of *BnaA07.MKK9$^{Hap1}$* and *BnaA07.MKK9$^{Hap2}$* were amplified from the Niklas and SWU47 accessions, respectively. After protein expression and purification, as described in the above section, 10 μg MBP (Sigma–Aldrich) were incubated with 20 μg GST-BnaA07.MKK9$^{Hap0}$ (corresponding to GST-BnaA07.MKK9, described above), BnaA07.MKK9$^{Hap1}$, or BnaA07.MKK9$^{Hap2}$ at 30 °C for 30 min in the kinase reaction buffer. Phosphorylation of MBP was detected by IB with the anti-pMBP antibody. Loading controls were detected with anti-GST and anti-MBP antibodies.

## MAPK activity assays

Tissues extending 10 mm beyond the inoculation site on the oilseed rape leaves were harvested at 0, 3, 6, 9, and 12 h after *S. sclerotiorum* inoculation. The intact *Arabidopsis* leaves were harvested 12 h after *S. sclerotiorum* inoculation. All samples were immediately stored at −80 °C. Total proteins were extracted in MAPK extraction buffer (100 mM HEPES, [pH 7.5], 5 mM EDTA, 5 mM EGTA, 10 mM Na$_3$VO$_4$, 10 mM NaF, 50 mM *β*-glycerophosphate, 10 mM DTT, 1 mM phenylmethylsulfonyl fluoride, 5 g mL$^{-1}$ leupeptin, 5 g mL$^{-1}$ aprotinin, 10% glycerol). The lysates were cleared by centrifugation at 15,800 × *g* for 30 min. Bradford assays were used to determine the protein concentrations of the supernatants[61]. Phosphorylated BnaMPK3, BnaMPK6, AtMPK3, and AtMPK6 were detected by IB with the anti-pTEpY antibody. The protein levels of BnaMPK3 and BnaMPK6 were detected by immunoblotting with the anti-BnaMPK3 and anti-BnaMPK6 antibodies, respectively, with Actin as the loading control.

## Immunocomplex kinase activity assays

To determine the kinase activity of BnaA07.MKK9 in the immunocomplex, MBP was used as substrate. The immunocomplex kinase activity assay was performed according to previous research[62] with some modifications. Leaf tissues extending 10 mm beyond the inoculation site were harvested at 12 h after *S. sclerotiorum* inoculation of oilseed rape leaves. Total proteins were extracted with immunoprecipitation buffer (50 mM Tris-HCl [pH 7.5], 150 mM NaCl, 5 mM EDTA, 5 mM EGTA, 5 mM Na$_3$VO$_4$, 5 mM NaF, 50 mM *β*-glycerophosphate, 10 mM DTT, 1 mM phenylmethylsulfonyl fluoride, 5 g mL$^{-1}$ leupeptin, 5 g mL$^{-1}$ aprotinin, 10% glycerol, 0.1% Tween 20) and centrifuged at 12,000 × *g* for 30 min at 4 °C. The 200 μg protein extracts were incubated with the anti-BnaA07.MKK9 antibody at 4 °C overnight and then incubated with BeyoMag™ protein A + G beads (Beyotime Biotech, China, P2108) at 4 °C for another 2 h. The beads retaining immunoprecipitated proteins were washed three times with

immunoprecipitation buffer and then incubated with 10 μg MBP in kinase buffer (25 mM Tris-HCl [pH 7.5], 5 mM MgCl$_2$, 1 mM EGTA, 0.1 mM Na$_3$VO$_4$, 1 mM DTT and 100 μM ATP) at 30 °C for 30 min. Phosphorylated MBP was analyzed by IB with the anti-pMBP antibody. The protein concentration of MBP was measured with the anti-MBP antibody.

## RNA extraction, RT-PCR and qRT-PCR assay

Total RNA was extracted using the RNAiso reagent kit (Vazyme) according to the manufacturer's protocol. One μg total RNA was reverse transcribed into first-strand cDNA using the HiScript II 1st Strand cDNA Synthesis Kit (Vazyme). RT-PCR was performed using Taq DNA polymerase with gene-specific primers. *AtActin2* (AT3G18780) and *BnaActin7* were used as the internal control. qRT-PCR assays were carried out using AceQ Universal SYBR qPCR Master Mix (Vazyme) on an ABI Step One Plus real-time PCR system (Applied Biosystems Inc., Foster City, CA, USA). *BnaUBC9* (BnaC08g12720D), *BnUBC10* (BnaA10g06670D), *AtUBQ10* (At5g53300) and *AtEF-1α* (AT5G60390) were used as standard controls.

## RNA-seq analysis

RNA-seq analysis of 4-week-old leaves of *Csv::BnaAO7.MKK9$^{DD}$ Arabidopsis* plants harvested at 2 h or 8 h after MOF or DMSO treatment was conducted for two independent biological replicates. *N. benthamiana* leaves were collected at 12 h after transient expression of *BnaAO7.MKK9$^{DD}$* or *BnaAO7.MKK9$^{KR}$* and three independent biological replicates were analyzed. As for oilseed rape, two independent biological replicates of *S. sclerotiorum* or mock-inoculated leaves were harvested at 12 hpi and subjected to RNA-seq analysis. The RNA libraries were constructed and sequenced on an Illumina Novaseq 6000 sequencer by Majorbio Company (Shanghai, China). The sequencing was performed as paired-end reads that were 2 × 150 bp in length. RNA-seq analysis was performed as previously described[63]. Raw data (raw reads) of fastq format were firstly subjected to quality control using FastQC (v.0.11.9). Then reads were mapped to the genome using HISAT2 (v.2.2.0) with default parameters. The *Arabidopsis* (TAIR10), oilseed rape (ZS11.v0), and *N. benthamiana* (v1.0.1) genomes served as reference genomes. DEGs were identified with a threshold of |log$_2$ fold changes| ≥ 1 and a false discovery rate (FDR) < 0.05. The significance of the GO and KEGG terms was corrected using FDR ≤ 0.05. A heatmap of the expression levels of the DEGs was constructed using TBtools[64].

## DAB staining

For staining, detached leaves were placed in a 3, 3'-diaminobenzidine (DAB) solution (1 mg mL$^{-1}$ DAB dissolved in ddH$_2$O, pH 3.8) overnight. The leaves were then transferred into 95% ethanol with gentle shaking at room temperature until entirely destained. H$_2$O$_2$ production was visualized as a reddish-brown coloration. Images were taken with a camera.

## Measurement of plant hormones, ACC, camalexin, H$_2$O$_2$, GSL

In *Arabidopsis*, the 4-week-old leaves of *Csv::BnaAO7.MKK9$^{DD}$* plants were harvested at 0, 2, and 8 h after MOF or DMSO treatment. In oilseed rape, tissues extending 10 mm beyond the inoculation site on the leaves were harvested at 0 hpi and 12 hpi after *S. sclerotiorum* or mock inoculation.

JA, JA-Ile, and SA were extracted following a previously described method[65]. Briefly, 100 mg lyophilized samples were extracted in extraction buffer (isopropanol: water: 37% HCl (2: 1: 0.002, v/v/v)) with shaking at 100 rpm at 4 °C for 30 min. One ml dichloromethane was added to each sample and shaken at 4 °C for 30 min. After centrifugation at 13,500 × g for 10 min, the supernatants were transferred to 1.5 mL microcentrifuge tubes and dried under a stream of nitrogen gas. Samples were redissolved in 400 μL methanol. JA, JA-Ile, and SA

contents were determined by HPLC-MS/MS analysis by a previously described method[66]. The elution profile was as follows: 0.1% formic acid in distilled water (v/v) and 0.1% formic acid in methanol (v/v) were used as mobile phases A and B. Apply a gradient as follows: 0–2 min, 0%–30% B; 2–20 min, 30%–100% B; 20–22 min, 100% B; and 22–25 min, 100–30% B. Standard solutions of JA, JA-Ile, and SA were analyzed to construct standard curves.

For ET production, the *Arabidopsis* leaves were detached, weighed, and placed on moist paper for 1 h at room temperature in the dark. Leaves were then transferred into 20 mL glass vials and immediately sealed with airtight Suba-Seal stoppers. After incubation in the dark for 12 h at room temperature, ET was measured by gas chromatography on an Agilent, 7890B system following a previously described method[67]. Temperatures for the injection port, column, and detector were kept constant at 140, 100, and 200 °C, respectively. Nitrogen was used as carrier gas at a flow rate of 30 mL min$^{-1}$, and hydrogen and air were used for the flame ionization detector at the rate of 30 mL min$^{-1}$, respectively. ACC was quantified as described by Cheng et al.[68]. The assay was based on the liberation of ethylene from ACC with NaOCl in the presence of Hg$^{2+}$. The ACC content of each sample was calibrated with the corresponding ET transformation rate of the sample.

The camalexin contents were determined with a F-4500 fluorescence spectrophotometer (Hitachi, Tokyo, Japan) as described by Yang et al.[40]. Values of fluorescence were measured with excitation light at 315 nm and emission light at 385 nm The standard curve was constructed with camalexin standard solutions (Sigma−Aldrich).

GSLs were extracted according to Tan et al.[69]. The samples were extracted in 90% methanol in distilled water. After centrifugation at 13,500 × g for 5 min, the supernatant was transferred to 5 mL column containing 1 ml DEAE Sephadex A-25 (Sigma−Aldrich). 100 μL 2 mg/mL sulphatase solution (Sigma−Aldrich) was added to desulphurize the GSL, and 100 μL 5 mM and 40 μL 5 mM 2-propenyl GSL (Sinigrin, Sigma−Aldrich) were used as the internal standards for oilseed rape and *Arabidopsis*, respectively. GSL contents were determined by HPLC on an Agilent 1260 Infinity II (Agilent, Technologies, Santa Clara, CA, USA) as previously described with some modifications[70]. Compounds were detected at 229 nm. Distilled water and 20% acetonitrile in distilled water (v/v) were used as mobile phases A and B, respectively. Apply a gradient as follows: 0–1 min, 0% B; 1–25 min, 0%–100% B; and 25–38 min, 100%–0% B. The mobile phase flow rate was 800 μL min$^{-1}$.

H$_2$O$_2$ contents were measured using a Hydrogen Peroxide Content Detection Kit (BC3595, Solarbio, Beijing, China) according to the manufacturer's instructions. The absorbance was recorded at 415 nm using a Tecan Infinite microplate reader (Spark M200, Tecan Austria GmbH, Grodig, Austria).

Three biological replicates were performed in JA, JA-Ile, SA, camalexin, GSL, and H$_2$O$_2$ content measurements, and four biological replicates were performed in ET production and ACC contents measurements.

## Statistical analyses

Differences between the two groups were assessed using a two-tailed Student's *t*-test performed in Microsoft Excel. For multiple comparisons, significance analysis was performed with one-way analysis of variance (ANOVA) followed by Tukey's honestly significant difference (HSD) post hoc test in SPSS statistical software (V16.0).

## Primers

All primers used in this study can be found in Supplementary Data 5.

## Reporting summary

Further information on research design is available in the Nature Portfolio Reporting Summary linked to this article.

## Data availability

The RNA-seq data generated in this study have been deposited in the National Center for Biotechnology Information Sequence Read Archive database under accessions PRJNA1040954, PRJNA1041073, PRJNA1041558. Source data are provided with this paper.

## Code availability

The code of all tools used for analyses in this paper is publicly available and has been presented in the Methods section. No custom codes were generated for the present study.

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

## Acknowledgements

We thank Prof. Dongtao Ren (China Agricultural University) and Prof. Juan Xu (Zhejiang University) for providing the Atmkk9 (SAIL_060_H06) and AtMPK3SR mutants, respectively. We are grateful to Prof. Mingyi Jiang (Nanjing Agricultural University) for phosphorylation assay assistance and for providing Co-IP-related plasmids. We are also grateful to Prof. Chunyu Zhang and Prof. Yongming Zhou (Huazhong Agricultural University) for providing the MOF-inducible expression vector and the association panel, respectively. This work was supported by the National Key Research and Development Program of China (2023YFF1000700 to Y.W.), the National Natural Science Foundation of China (32272112 to J.W., 32172019 to Y.W.), the Natural Science Foundation of Jiangsu Province (BE2022340 to Y.W.), the Open Project of Key Laboratory of Biology and Genetic Improvement of Oil Crops, Ministry of Agriculture and Rural Affairs, China (KF2022007 to J.W.), the China Postdoctoral Science Foundation (2022M722703 to L.L.), the Priority Academic Program Development of Jiangsu Higher Education Institutions, the Jiangsu Qinglan Project (to J.W.) and the Jiangsu Funding Program for Excellent Postdoctoral Talent (2023ZB524 to L.L.).

## Author contributions

L.L. and J.W. designed the experiments. L.L., X.Z., and J.F. performed most of the experiments. J. L. and Q.-Y.Y. performed the GWAS analysis. S.R., X.G., M.X., and P.L. performed genetic transformation. W.L., J.X., and D.L. performed the SSR resistance assays. Q.S., G.C., and D.L. analyzed the data. L.L. and J.W. wrote the manuscript with comments from all authors. Y.W. and J.W. revised the manuscript and gave suggestions. All authors read and approved the final manuscript.

## Competing interests

The authors declare no competing interests.
