## [Peer Review File · Nature Communications]

Natural variation in BnaA07.MKK9 confers resistance to Sclerotinia stem rot in oilseed rapeREVIEWER COMMENTS

Reviewer #1 (Remarks to the Author):

Sclerotinia stem rot is a major disease of oilseed rape, leading to substantial reductions in both yield and quality. Currently, there is a scarcity of Sclerotinia-resistance genes available for breeding purposes. Lin et al. successfully identified a Sclerotinia stem rot resistance gene, BnaA07.MKK9, through a combination of GWAS and reverse genetics analyses. The authors elucidated the natural variation present in BnaA07.MKK9, thereby providing a valuable genetic resource for resistance breeding. They uncovered the significant role of the MKK9-MPK3/6 cascade in resistance to the necrotrophic pathogen in Arabidopsis and oilseed rape. The work is impressive and with a significant workload, due to the fact that oilseed rape is an allopolyploid, making the task of conducting comprehensive gene functional analysis challenging. I have a few comments that I would like to see addressed.

Major comments:

1. Constitutive overexpressing of BnaA07.MKK9DD resulted in severe growth defects in rapeseed. In Arabidopsis, some MAPK null mutants, such as Atmkk4/5 double mutant, Atmpk3/6 double mutant were lethal in seedling, suggesting their essential roles in plant development. How about the Bnamkk9 quadruple mutants?
2. The authors designed four gRNAs for editing of BnaMKK9. The gRNA1 and gRNA2 for BnaA02.MKK9 and BnaC02.MKK9. The gRNA3 and gRNA4 for BnaA07.MKK9 and BnaC06.MKK9. The editing events were identified in Tgt1 and Tgt3 of the four homologs in two quadruple mutants (#5 and #7). However, there is no information provided regarding Tgt2 and Tgt4.
3. The detection of potential off-target sites at each sgRNA target site is essential for the two Bnamkk9 quadruple mutants (#5 and #7).
4. Line 250-255, why the authors used kinase-inactive forms of BnaC03.MPK3 and BnaC03.MPK6?
5. Line 170, what do the authors mean by “these effects are specifically linked to the kinase activity of BnaA07.MKK9 and hypersensitive response (HR) cell death”? The results can’t support this conclusion.
6. In figure legend of Figure 4c-d, the interaction between BnaA07.MKK9KR and BnaC03.MPK3/6 was confirmed by Co-IP. But Figure 4c and Figure 4d were all showed the interaction between BnaA07.MKK9KR and BnaC03.MPK3. Furthermore, considering that both BnaMPK3 and BnaMPK6 have multiple copies in oilseed rape, why did the authors select BnaC03.MPK3 and BnaC03.MPK6 for analysis?
7. In the Supplementary Figure 11a-b, the expression levels of AtMPK3 and AtMPK6 in Col-0, Atmpk3 and Atmpk6 mutants were identified by RT-PCR, respectively. The authors successfully

detected the expression of AtMPK3 and AtMPK6 in the corresponding mutants. In contrast, in the WT, the expression of AtMPK3 and AtMPK6 was not observed. How do the authors explain this?

Minor comments:

1. Line 164, “BnaA07.MKK9WT” appears for the first time in the manuscript, but does not describe the method by which the material was obtained. Is it a generic overexpressed line driven by a 35S promoter? Or a wild type plant?
2. What is the difference between BnaMKK9DD and BnaA07.MKK9DD? Many parts of the plant materials, gene names and expressions are irregular in the whole manuscript, please revise.
3. It is recommended to add a clear icon in Fig.2m to indicate that the upper part is not infected with *S. sclerotiorum*.
4. Figure 6c, Hap-0, Hap-1 and Hap-2 should be replaced with Hap0, Hap1 and Hap2, respectively.
5. Supplementary Figure 13, there are eight samples on the bottom panel of the control protein. But in the top and middle panels, there are only six samples. How could the author make this mistake?
6. Line 164, “BnaA07.MKK9WT” appears for the first time in the manuscript, but does not describe the method by which the material was obtained. Is it a generic overexpressed line driven by a 35S promoter? Or a wild type plant?
7. For resistance assessment, n represents the number of plants in the experiment., but it does not represent biologically independent samples. Correct it in several figure legends.
8. The manuscript is quite well written, but there are some errors involving words in italics throughout the manuscript that require correction. For example, line 155 “T5”, line 207“(l3G) and”, line 217 “(ACC, an ET precursor)”, line 268 “and”
9. Line 271 “we increased their expression in J9712 using the CaMV 35s promoter.” This sentence is not accurate. Change to “we overexpressed.....”.
10. Figure 4, Clarify the meaning of Hr+S.s.
11. The production of protein-related figures in Figure 4c to 4i are not standardized, there are no units, the lanes are not aligned, and the wrong results are displayed. Please review the results and re-draw.
12. Fig. 6b, f, g, h, please redraw the figures to be consistent with the main color of the manuscript.
13. Line 653, Log2 fold changes $|\geq 1|$ should be $|\text{Log}_2 \text{ fold changes}| \geq 1$

Reviewer #2 (Remarks to the Author):

This paper presents a genome-wide association study analysis aimed at identifying the key gene responsible for the Sclerotinia stem rot resistance in rapeseed. The authors carry out a wide range

of interesting analyses to show that MKK9-MPK3/6 cascade involved in plant defense response to *S. sclerotiorum*, and the natural variation in BnaA07.MKK9 confers *Sclerotinia* resistance in rapeseed. These findings enhance our understanding of *Sclerotinia* stem rot resistance and offer a valuable genetic resource for the rapeseed breeding. The results are novel and original, the experimental design and data analysis of this study are reasonable, with clear logic and sufficient evidence, and the experimental results can support the conclusion of this paper.

The manuscript is well written, making it easily to follow. But I have the following comments and suggestions for the authors' consideration.

- (1) Due to the embryo lethality of the Atmpk3 Atmpk6 double mutant, the authors used a chemically-rescued variant, AtMPK3SR, which mimics the double mutant in the presence of the inhibitor NA-PP1. How the authors ensure the dual-blocking effect is achieved after the application of NA-PP1.
- (2) The authors employed quadruple mutants of BnaMKK9, but the single mutant, particularly the BnaA07.MKK9 knock out mutants were not found in the manuscript.
- (3) Why did the authors use three OE lines of BnaA07.MKK9 in the T1 generation (Fig.1a-d), but only two lines in the T2 generation (Supplementary Fig. 1)?
- (4) The author did not conduct off-target analysis during gene editing (Fig 2e). How did they determined that the phenotype resulted from the edits of BnaMKK9?
- (5) The gel images should be aligned in Fig.4g (the anti-GST).
- (6) The author should provide clarification regarding the BnaA07.MKK9 haplotype of transgenic receptor J9712.
- (7) For qRT-PCR, the author utilized only one reference gene. It is recommended to employ at least two validated reference genes.
- (8) Figure 4a: Why used the kinase active variant, kinase inactive variant and wild type of BnaA07.MKK9 to test the interaction of BnaC03.MPK3 and BnaC03.MPK6 by two-hybrid analysis? Only weak interaction was found between BnaA07.MKK9KR and BnaC03.MPK3/6. I recommend that the author provides possible explanations for this observation.
- (9) Does the author conduct molecular detection for the Arabidopsis mutants before phenotypic analysis, such as Atmkk9-1 (SALK_017378) and Atmkk9-2 (SAIL_60_H06)?
- (10) The numbers on the protein gel images lack units, kilodaltons?

Reviewer #3 (Remarks to the Author):

This paper reports cloning of a gene conferring what appears to be a strong resistance against *Sclerotinia sclerotiorum* in oilseed rape/rapeseed/canola (*Brassica napus*), one of the most important global oilseed crops. Stem rot caused by *Sclerotinia* is one of the most widespread, important and challenging diseases of rapeseed and canola, with very high importance in the three

most important growing areas (China, Europe and North America). However, breeding for resistance remains challenging due to low heritability of resistance/susceptibility and a lack of knowledge about effective, quantitative resistance loci. Sclerotinia disease is also important in many of the world's other major oilseed, protein and vegetable crops (e.g. soybean, chickpea and other legumes, sunflower, potato and several vegetable species), however to date there has also been little progress in implementation of effective genetic resistance in other crops.

Hence, the present paper could represent an important breakthrough. I think the authors have done a very good job of identifying and characterising the resistance response activated by BnaA07.MKK9, so the work could become a catalyst for biotechnological approaches to raise resistance in *B. napus* and other crop species.

I am not an expert on plant-pathogen interactions or disease resistance mechanisms, hence I will limit my comments mainly to genetic aspects of the gene identification and validation, and the practical implications. Overall I think this is a clearly-formulated and coherent manuscript which sufficient value for publication. However I have major questions regarding the interpretation of the results and the practical relevance of this particular gene in oilseed rape breeding.

Major comments:

The GWAS was performed using phenotype data from 322 rapeseed accessions, Please add details in the first paragraph of the Methods section describing the rationale for assembly of this panel (particularly what is known about the diversity of the panel in relation to the species-wide diversity of *B. napus* which is reported elsewhere in the paper). In particular, I am interested to know how the frequency of the resistance alleles in the GWAS panel reflects the bias found in the global *B. napus* panel, where the resistant haplotype Hap0 was subsequently found to be dominant in European winter and Asian semi-winter gene pools, though not so widespread in North American spring-type canola. To my knowledge no previous GWAS study of Sclerotinia resistance in a randomly assembled, genetically diverse population has revealed such a clear, major-effect QTL with high additive effects. Hence this result suggests that the GWAS panel has a very clear segregation for this major-gene effect that is not normally the case in (randomly selected) *B. napus* diversity panels. Indeed, the data in Figure 6 show that susceptible haplotypes are rare in this species. Hence, one gains the impression that the paper is actually reporting cloning of a relatively rare susceptibility locus. In this case the results would have much less practical relevance for breeding than the manuscript suggests.

Figure 6 suggests that the resistance haplotype Hap0 is widespread in European winter and Asian semi-winter accessions, but less so in North American spring-type accessions. It further reports that the frequency of Hap0 was always very high, however it decreased in the period when global

canola/oilseed rape breeding was focussed on conversion to 00 seed quality, and increased again in recent decades. What the authors do not show is how frequent Hap0 is in current spring-type, 00 canola cultivars in North America. I would suspect that the frequency is also very high, given that Canadian breeders have been selecting for quantitative Sclerotinia resistance for decades, so that a major-effect locus like this one would normally become quickly fixed for positive haplotypes through simple phenotypic selection. If this is not the case, the simplest explanation would be that the resistance is not effective against fungal pathotypes that are important in practical cropping systems in Canada. How many of the Hap1 and Hap2 genotypes released since 2000 (right-hand diagram in Figure 6h) are North American/European spring-type cultivars, how many are European winter-type, and how many are Asian semi-winter type?

In regard to practical relevance of the results, on line 422 the authors state that field experiments included two replications for the association panel, with 6-8 plants per accession being artificially inoculated on the stems with a single pathogen isolate (SS-1). Given that Sclerotinia infection typically has a very high environmental variance and G*E interaction (even with "controlled" stem inoculations, which are also not highly repeatable), I am concerned that there appears to be no validation in an independent field environment and/or with different isolates. Hence, it is not demonstrated that plants carrying the major resistance QTL against SS-1 are truly more resistant under natural Sclerotinia infection (including challenge by potentially multiple isolates) in different field environments. In the introduction, the authors emphasise the extreme damage caused by Sclerotinia in China and elsewhere (lines 48-52). If the resistance is as effective as the manuscript suggests, how can the high frequency of resistance haplotype Hap0 in Chinese and European cultivars be reconciled with extremely high infection levels in farmer's fields?

I also find it unusual that a leaf assay was used alongside stem inoculations to screen resistance in transgenic plants, but not in the association panel. To my knowledge and from personal experience, the transferability of resistance/susceptibility scores between leaf and stem assays is generally very poor (even if it may correlate in specific, extreme genotypes with specific isolates). Indeed, Brassica pathologists are still very divided about which (if any) of these methods gives truly relevant results for natural Sclerotinia infections in a field cropping situation. Most agree that an isolate mixture (rather than single isolates) come closer to a natural infection situation, although individual isolates are important for detailed characterisation of plant responses. However, in the transgenic plants there seems to be a high convergence of the disease scores. If this simple assay discriminates so well for this gene, is there a plausible reason why the authors did not support their stem inoculation data in the GWAS with additional leaf inoculation data? Also, why were Hap0-carrying accessions not challenged with other pathogen isolates?

Furthermore, no information is provided on the heritability of the infection response, which is generally very low for Sclerotinia disease in field situations. Since plants for the GWAS were grown in only a single environment, the authors can only calculate h^2 across the two replications (i.e.

“repeatability”) as a proxy for heritability. At least that might improve confidence in the reliability of the phenotype data (with this single isolate) in the absence of multiple field environments.

Overall, I think the manuscript reports cloning of an interesting susceptibility gene to a major disease in a major crop, which is a novel finding. However, the resistant haplotype of this gene is already widespread in major oilseed rape/canola cultivars (and I would be rather surprised if it is not more or less fixed in the most recent, high-yielding 00-type cultivars in all ecotype groups). Hence, more detailed results and explanations are needed to convince me that the gene has a real practical relevance for breeding that has not already been exploited by simple phenotypic selection for disease scores and yield performance in modern cultivars. This is a solid and well-written manuscript but in light of these concerns I consider the main message and conclusions to be somewhat exaggerated.

Minor comments:

Lines 33-34: “Rapeseed (*Brassica napus*)” – I recommend to refer throughout the manuscript to “oilseed rape” rather than rapeseed. This is necessary to distinguish 00-quality oilseed rape/canola types (the form that is today grown globally) from old ++ or +0 types which are barely grown today (internationally commonly referred to as rapeseed in English). To emphasise the importance of the findings for all common *B. napus* crop types, I recommend to change this particular phrase to “Oilseed rape (*Brassica napus*), also known as canola or rapeseed, is a globally important edible oil crop that emerged...”

Lines 33-34: “...around 7,500 years ago ...”

The supposedly exact timing of the origin of *B. napus* 7,500 years ago is rather controversial and I personally prefer to avoid using this explicit date. In the genome assembly paper of Chalhoub et al. (2014) in *Science*, this wording was introduced in the abstract of the manuscript by the journal editors, although the authors actually stressed that the evidence from the “molecular clock” data placed the origin of *B. napus* at “no older than 7,500 years”. As a consequence, this date is frequently misconstrued as fact: However if you read Chalhoub et al. (2014) carefully, the main manuscript text states that *B. napus* formed after this date. As noted in the present manuscript on line 37, the first indications for *B. napus* cultivation arose in the 13th Century and there is no evidence that the species arose earlier. Since no natural forms have ever been found, the available evidence suggests that the time of origin was considerably more recent than the “maximum” 7,500 years ago. I would prefer if you change your statement to “a globally important edible oil crop that emerged during the past 7,500 years”.

Line 34: Please change “from a natural hybridization event” to “from natural hybridisation”. To my knowledge it is unclear whether there was one or several hybridisation events, but the latter appears more likely.

Lines 41-44: “Double low breeding technique” is not a technique for breeding, rather double low cultivars are simply the result of classical breeding implementing chemical analysis for seed quality selection. Please change to “Over the past five decades, double-low cultivars with low seed erucic acid and glucosinolate content, pioneered in Canada, significantly improved B. napus oil and meal quality.”

Line 83: “we identified “a particular gene”, BnaA07.MKK9, as a significant factor in SSR resistance in rapeseed. Please change to “we identified the MKK gene BnaA07.MKK9 as a significant factor contributing to SSR resistance in rapeseed.

Point-by-point response to the reviewers' comments

Dear Reviewers,

Thank you all for your invaluable comments and suggestions regarding our manuscript, which have significantly enhanced the quality of our study. In response to your concerns, we have provided additional data and meticulously revised our manuscript. Please find our point-by-point response to your comments below. For clarity, our responses are written in blue.

Responses to comments by Reviewer #1:

Sclerotinia stem rot is a major disease of oilseed rape, leading to substantial reductions in both yield and quality. Currently, there is a scarcity of Sclerotinia-resistance genes available for breeding purposes. Lin et al. successfully identified a Sclerotinia stem rot resistance gene, *BnaA07.MKK9*, through a combination of GWAS and reverse genetics analyses. The authors elucidated the natural variation present in *BnaA07.MKK9*, thereby providing a valuable genetic resource for resistance breeding. They uncovered the significant role of the MKK9-MPK3/6 cascade in resistance to the necrotrophic pathogen in Arabidopsis and oilseed rape. The work is impressive and with a significant workload, due to the fact that oilseed rape is an allopolyploid, making the task of conducting comprehensive gene functional analysis challenging. I have a few comments that I would like to see addressed.

Response:

We deeply appreciate the reviewer for recognizing our hard work, and for helping us improve our manuscript.

Major comments:

1. Constitutive overexpressing of *BnaA07.MKK9DD* resulted in severe growth defects in rapeseed. In Arabidopsis, some MAPK null mutants, such as *Atmkk4/5* double mutant, *Atmpk3/6* double mutant were lethal in seedling, suggesting their essential roles in plant development. How about the *Bnamkk9* quadruple mutants?

Response:

Many thanks for the helpful comment. Indeed, we had previously examined the primary agronomic traits of J9712 and *Bnamkk9* quadruple mutants. The major agronomic traits of *Bnamkk9* quadruple mutants showed no significant alterations under field conditions. We have added this result to Supplementary Table 4, and the Results section (lines 169-171) of the revised manuscript.

Supplementary Table 4 Agronomic traits of the J9712 and *Bnamkk9* mutants in the experimental field at Yangzhou University

Line	Plant height (cm)	Branch initiation height (cm)	First effective branch number	Siliques number per plant	Siliques length (cm)	Siliques seed number	Thousand-seed weight (g)
J9712	167.38±3.56	63.99±2.04	9.27±1.03	340.20±16.88	10.55±0.44	20.80±1.01	4.27±0.05
Bnamkk9 -#5	168.08±4.43	64.77±2.90	9.08±1.16	332.17±20.59	10.59±0.56	20.92±1.16	4.30±0.08
Bnamkk9 -#7	164.46±3.56	62.68±2.57	9.46±1.13	327.69±16.75	10.44±0.44	20.69±1.18	4.31±0.06

Note: Values are means ± SD. A total of 13-15 plants were randomly selected for each line.

2. The authors designed four gRNAs for editing of BnaMKK9. The gRNA1 and gRNA2 for BnaA02.MKK9 and BnaC02.MKK9. The gRNA3 and gRNA4 for BnaA07.MKK9 and BnaC06.MKK9. The editing events were identified in Tgt1 and Tgt3 of the four homologs in two quadruple mutants (#5 and #7). However, there is no information provided regarding Tgt2 and Tgt4.

Response:

Many thanks for the helpful suggestion, we have analyzed sequencing data of Tgt2 and Tgt4 in two *Bnamkk9* quadruple mutants (#5 and #7). However, no editing events were identified. We have added these findings to Supplementary Table 2 of the revised manuscript.

3. The detection of potential off-target sites at each sgRNA target site is essential for the two *Bnamkk9* quadruple mutants (#5 and #7).

Response:

Thank you very much for your comment, which led us to search for potential off-target sites in the CRISPR-P 2.0 database (<http://crispr.hzau.edu.cn/cgi-bin/CRISPR2/>). A total of 48 putative off-target sites were identified for the 4 sgRNAs, and no off-target editing was detected in *Bnamkk9* quadruple mutants (#5 and #7). We have added these findings to the Results section (lines 161-163), the Methods section (lines 503-506) and Supplementary Table 3 in the revised manuscript. Additionally, specific primers have been included in Supplementary Table 3.

Supplementary Table 3 Detection of potential off-target sites at each sgRNA target site of *Bnamkk9* quadruple mutants (#5 and #7) in *T₄* progenies

Sequence name	Sequence	No. of mismatching bases	Putative off-target locus in Darmor reference genome	Region	No. of lines sequenced (T4)	No. of mutations
BnaMKK9-Tgt1						
	GAGCGTAGAGAGTGTCTTGTGG		BnaA02.MKK9/BnaC02.MKK9			
Off-1	GAA C ATAGAGAA T GTCTT A AGG	4	BnaA09g47620D	CDS	2	0
Off-2	GAA C ATAGAGAA T GTCTT A AGG	4	BnaC08g41990D	CDS	2	0
Off-3	GG C CAAGAGAGTATCTT G AGG	4	BnaC02g34940D	CDS	2	0
Off-4	GG C CAAGAGAGTATCTT G AGG	4	BnaA09g18900D	CDS	2	0
Off-5	TT C GTAGAGAGTGTCTT G AGG	4	BnaCng08990D	CDS	2	0
Off-6	GAGTGT G AA T GTGTCTT G GGG	4	BnaA09g50420D	CDS	2	0
Off-7	GAGCT C ACAG A TGTCTT C GGG	4	BnaC03g06070D	CDS	2	0
Off-8	GAGCG G AG G GAGTCTT C TATGG	4	BnaC02g00110D	CDS	2	0
Off-9	GAGCG G AG G GAGTCTT C TATGG	4	BnaA02g00690D	CDS	2	0
Off-10	GAGT T TAGAG A TGA T CTT G AGG	4	chrC06:-2219831	Intergenic	2	0
Off-11	GAGCT T GG G AGT G ATCTT G GGG	4	BnaA06g13840D	CDS	2	0
Off-12	GAGCT T GG G AGT G ATCTT G GGG	4	BnaC05g15270D	CDS	2	0
Off-13	GACCGTAGAGAG C TTTCTT G AGG	3	chrCnn_random:+57263323	Intergenic	2	0
Off-14	GAGC S T A AGAG S GTCTT C T G GGG	4	chrC06:+3916958	Intergenic	2	0
Off-15	GAGCG T AGAG G GTTCTT C GGG	4	chrC01:+36276132	Intergenic	2	0
BnaMKK9-Tgt2						
	CATCCTTCGGCGCTCGCGCGGG		BnaA02.MKK9/BnaC02.MKK9			
Off-16	CATC C CT C AG C GC T TCAG C CGG	4	chrA10:-4096222	Intergenic	2	0
Off-17	TATC T TCGG T GC T CGC T CGG	4	chrC06:-434952	Intergenic	2	0
Off-18	CATA C CT C CGC C CGC C CGCGCGG	4	BnaC06g23550D	CDS	2	0
Off-19	CATC T CG C CG C CGC T CGCGCGG	4	BnaC03g36200D	CDS	2	0
BnaMKK9-Tgt3						
	ACGGCAGCTGATGCGAGAGATGG		BnaA07.MKK9/BnaC06.MKK9			
Off-20	ATGGC A CT T GG T AC G AGAGAA G	4	chrC07:+9916222	Intergenic	2	0
Off-21	ATGGC A CT T GG T AC G AGAGAA G	4	chrA02:-4217962	Intergenic	2	0
Off-22	TCGG A AG T AG T GT G AGAGAC G	4	BnaC04g43320D	CDS	2	0
Off-23	TCGG C GG T GT A AGAGAGAA G	4	BnaC04g25460D	CDS	2	0
Off-24	TCGG C GG T GT A AGAGAGAA G	4	BnaA04g03570D	CDS	2	0
Off-25	ACGG A AG T GT A T G AGAGAG A G	3	BnaCng19760D	CDS	2	0
Off-26	ACAG G AG C TT A AGC G AGAG A TAG	4	BnaC05g18220D	CDS	2	0
Off-27	ACAG G AG C TT A AGC G AGAG A TAG	4	BnaA09g30260D	CDS	2	0
Off-28	ACGG C AG T GT A T G CG G AG T GGG	4	BnaA08g11990D	CDS	2	0
Off-29	ACGG C AG T GT A T G CG G AG T GGG	4	BnaC03g66880D	CDS	2	0
Off-30	ACGG C AG T GT A T G CG A AG A ATAG	4	BnaA02g11830D	CDS	2	0
Off-31	CGGG A GC T GT A T G CG G AG A TAG	4	BnaA01g31990D	CDS	2	0
BnaMKK9-Tgt4						
	CCTCCTCAACTCGAAGGACGAGG		BnaA07.MKK9/BnaC06.MKK9			
Off-32	A CTC T T A ACT C G A AG G AC G AG G	3	BnaC02g22290D	CDS	2	0
Off-33	TCTC T CA A G T CG A AG A CT C GG	4	chrC03:-5468084	Intergenic	2	0
Off-34	CCT T CA A CT C CA A AG A CT G GG	4	BnaC08g35380D	CDS	2	0
Off-35	CCTC T CA A CT C CA A AG A CT C GG	4	chrA07:-2287937	Intergenic	2	0
Off-36	CCT T CT C CA A CT C AG A GG A CG A GG	4	chrC08:+691761	Intergenic	2	0
Off-37	CCT T CT C CA A CT C AG A GG A CG A GG	4	chrCnn_random:-50767481	Intergenic	2	0
Off-38	CCTC T CA A CT C CA A AG A CT G GG	4	BnaC05g44450D	CDS	2	0
Off-39	CCTC T CA A CT C CA A AG A CT G GG	4	BnaA05g30120D	CDS	2	0
Off-40	CCT G AT C CA A CT C CA A AG A CG A GG	3	chrAnn_random:+9762623	Intergenic	2	0
Off-41	CCT G CT A ACT C CA A AG A T G AGG	4	chrAnn_random:+18909459	Intergenic	2	0
Off-42	CCTC G CA A CT C CA A AG A CG A GG	4	BnaAnn20150D	CDS	2	0
Off-43	CCT G AT C ACT C CA A AG A CG A GG	4	chrAnn_random:+23122088	Intergenic	2	0
Off-44	CCT G AT C ACT C CA A AG A CG A GG	4	chrAnn_random:+37208866	Intergenic	2	0
Off-45	CCT G AT C ACT C CA A AG A CG A GG	4	chrAnn_random:-45807849	Intergenic	2	0
Off-46	CCT G AT C ACT C CA A AG A CG A GG	4	chrAnn_random:-41533491	Intergenic	2	0
Off-47	CCT G AT C ACT C CA A AG A CG A GG	4	chrAnn_random:-41441798	Intergenic	2	0
Off-48	CCT G AT C ACT C CA A AG A CG A GG	4	chrAnn_random:-15066110	Intergenic	2	0

The protospacer adjacent motif (PAM) is highlighted in green color. The mismatched nucleotides of each putative off-target sites are shown in red color.

4. Line 250-255, why the authors used kinase-inactive forms of BnaC03.MPK3 and BnaC03.MPK6?

Response:

Thank you for this insightful comment. MPK3 and MPK6 had autophosphorylation activity, therefore it was not an ideal option to detect the phosphorylation by MAPKKs (Yang et al., 2001; Xu et al., 2018). BnaC03.MPK3^{KR} and BnaC03.MPK6^{KR}, the kinase-inactive forms of BnaC03.MPK3 and BnaC03.MPK6, lacked autophosphorylation activity (Fig. 4g, h), making it easy to detect phosphorylation by BnaMAPKKs. Hence, we used the kinase-inactive forms of BnaC03.MPK3 and BnaC03.MPK6 to detect their phosphorylation by BnaA07.MKK9.

References:

Xu, R. et al. Control of grain size and weight by the OsMKKK10-OsMKK4-OsMAPK6 signaling pathway in rice. *Mol. Plant*, **11**, 860-873 (2018).

Yang K., Liu Y. & Zhang S. Activation of a mitogen-activated protein kinase pathway is involved in disease resistance in tobacco. *Proc. Natl Acad. Sci. USA*, **98**, 741-746 (2001).

5. Line 170, what do the authors mean by “these effects are specifically linked to the kinase activity of BnaA07.MKK9 and hypersensitive response (HR) cell death”? The results can’t support this conclusion.

Response:

Many thanks for pointing this out. Following your comment, we have revised the sentence in question to “implying a correlation between the kinase activity of BnaA07.MKK9 and hypersensitive response (HR) cell death” (Lines 183-184) in the revised manuscript.

6. In figure legend of Figure 4c-d, the interaction between BnaA07.MKK9KR and BnaC03.MPK3/6 was confirmed by Co-IP. But Figure 4c and Figure 4d were all showed the interaction between BnaA07.MKK9KR and BnaC03.MPK3. Furthermore, considering that both BnaMPK3 and BnaMPK6 have multiple copies in oilseed rape, why did the authors select BnaC03.MPK3 and BnaC03.MPK6 for analysis?

Response:

Thank you for this astute observation. This was an error on our part, and we have corrected this mistake in the revised manuscript. The interaction between BnaA07.MKK9^{KR} and BnaC03.MPK3 is presented in Fig. 4c, while the interaction between BnaA07.MKK9^{KR} and BnaC03.MPK6 is shown in Fig. 4d.

Based on the BnIR database (<https://yanglab.hzau.edu.cn/>), *BnaMPK3* has two homologs, namely *BnaA06.MPK3* (BnaA06G0184800ZS) and *BnaC03.MPK3* (BnaC03G0606500ZS). For *BnaMPK6*, there are four homologs: *BnaC03.MPK6* (BnaC03G0251900ZS), *BnaC04.MPK6* (BnaC04G0044200ZS), *BnaA03.MPK6* (BnaA03G0213400ZS) and *BnaA05.MPK6* (BnaA05G0040500ZS). Given the remarkable degree of amino acid homology among these homologous genes, we chose

to select only a single copy for our analysis of interactions. In our earlier study, we conducted transcriptomic analyses of *B. napus* before and after *S. sclerotiorum* infection. *BnaC03.MPK3* and *BnaC03.MPK6* were the most strongly induced homologs post infection (Wu et al., 2016). As a result, we selected *BnaC03.MPK3* and *BnaC03.MPK6* for the interaction analysis.

Reference:

Wu, J. et al. Comparative transcriptomic analysis uncovers the complex genetic network for resistance to *Sclerotinia sclerotiorum* in *Brassica napus*. *Sci. Rep.* **6**, 19007 (2016).

7. In the Supplementary Figure 11a-b, the expression levels of AtMPK3 and AtMPK6 in Col-0, *Atmpk3* and *Atmpk6* mutants were identified by RT-PCR, respectively. The authors successfully detected the expression of AtMPK3 and AtMPK6 in the corresponding mutants. In contrast, in the WT, the expression of AtMPK3 and AtMPK6 was not observed. How do the authors explain this?

Response:

Thank you for pointing this out, we apologize for any confusion caused. We mislabeled Supplementary Figure 12a-b (Supplementary Figure 11a-b in the previously version). In Supplementary Figure 12a, we mistakenly reversed the labels for *Atmpk3* and WT. Similarly, in Supplementary Figure 12b, we mislabeled *Atmpk6* and WT. We conducted additional experiments which confirmed that the labeling was indeed reversed. We have updated Supplementary Figure 12 in the revised manuscript.

Minor comments:

1. Line 164, “*BnaA07.MKK9WT*” appears for the first time in the manuscript, but does not describe the method by which the material was obtained. Is it a generic overexpressed line driven by a 35S promoter? Or a wild type plant?

Response:

Thank you for pointing this out. We transiently expressed both *BnaA07.MKK9^{WT}* (representing the unmodified wild-type gene) and *BnaA07.MKK9^{DD}* in *N. benthamiana* leaves. It is important to note that "WT" does not refer to the transgenic or wild-type plant in this context. To prevent any potential confusion, we have revised the manuscript by changing “*BnaA07.MKK9^{WT}*” to “*BnaA07.MKK9*”.

2. What is the difference between *BnaMKK9DD* and *BnaA07.MKK9DD*? Many parts of the plant materials, gene names and expressions are irregular in the whole manuscript, please revise.

Response:

Thank you for pointing this out, we apologize for the confusion. *BnaMKK9^{DD}* was the abbreviation of *BnaA07.MKK9^{DD}* in one of our previous manuscripts. In the revised manuscript, all mentions of *BnaMKK9^{DD}* have been corrected to *BnaA07.MKK9^{DD}*.

3. It is recommended to add a clear icon in Fig.2m to indicate that the upper part is not infected with *S. sclerotiorum*.

Response:

Thank you for your careful review. We have added a clear icon in Figure 2m in the revised manuscript.

4. Figure 6c, Hap-0, Hap-1 and Hap-2 should be replaced with Hap0, Hap1 and Hap2, respectively.

Response:

Thank you for your careful review. Following your comment, we have replaced them in the revised manuscript.

5. Supplementary Figure 13, there are eight samples on the bottom panel of the control protein. But in the top and middle panels, there are only six samples. How could the author make this mistake?

Response:

We apologize for our mistake, and have corrected this in Supplementary Figure 15 (Supplementary Figure 13 in the previously version) of the revised manuscript.

6. Line 164, “BnaA07.MKK9WT” appears for the first time in the manuscript, but does not describe the method by which the material was obtained. Is it a generic overexpressed line driven by a 35S promoter? Or a wild type plant?

Response:

Thank you. We have addressed this issue in Minor comment 1.

7. For resistance assessment, n represents the number of plants in the experiment.", but it does not represent biologically independent samples. Correct it in several figure legends.

Response:

Thank you for noting this issue. We have modified the description in the Figure Legends (lines 808, 830 and 877) and Supplementary Figure legends in the revised manuscript.

8. The manuscript is quite well written, but there are some errors involving words in italics throughout the manuscript that require correction. For example, line 155 “T5”, line 207“(I3G) and”, line 217 “(ACC, an ET precursor)”, line 268 “and”

Response:

We appreciate your thorough review. Regarding the issues with words in italics, we have carefully addressed and corrected them throughout the entire manuscript.

9. Line 271 “we increased their expression in J9712 using the CaMV 35s promoter.” This sentence is not accurate. Change to “we overexpressed.....”.

Response:

Thanks for your advice. According to the reviewer’s comment, we have revised the sentences. The updated sentence now reads: “we overexpressed *BnaC03.MPK3* and *BnaC03.MPK6* in J9712 using the CaMV 35S promoter”.

10. Figure 4, Clarify the meaning of Hr+S.s.

Response:

Thank you for helping us improve the clarity of our writing. In Figure 4i, “Hr + S.s” indicates the time in hours at which inoculation with *S. sclerotiorum* was conducted. According to your suggestion, we have added this description in the figure legend of Figure 4i (line 859).

11. The production of protein-related figures in Figure 4c to 4i are not standardized, there are no units, the lanes are not aligned, and the wrong results are displayed. Please review the results and re-draw.

Response:

Many thanks for your careful reading. We redrew the results in Figure 4 from Figure 4c to Figure 4i in the revised manuscript. The protein units were added in figure legends (lines 784, 860 and 898) and Supplementary figure legends.

12. Fig. 6b, f, g, h, please redraw the figures to be consistent with the main color of the manuscript.

Response:

Thank you, we adjusted the color of Figure 6b, 6f, 6g and 6h in the revised manuscript.

13. Line 653, Log2 fold changes $|\geq 1|$ should be $|\text{Log}_2 \text{ fold changes}| \geq 1$

Response:

Thank you for the great suggestion. We have replaced Log2 fold changes to $|\geq 1|$ to $|\text{Log}_2 \text{ fold changes}| \geq 1$ in line 694 of the revised manuscript.

Responses to comments by Reviewer #2:

This paper presents a genome-wide association study analysis aimed at identifying the key gene responsible for the *Sclerotinia* stem rot resistance in rapeseed. The authors carry out a wide range of interesting analyses to show that MKK9-MPK3/6 cascade involved in plant defense response to *S. sclerotiorum*, and the natural variation in *BnaA07.MKK9* confers *Sclerotinia* resistance in rapeseed. These findings enhance our understanding of *Sclerotinia* stem rot resistance and offer a valuable genetic resource for the rapeseed breeding. The results are novel and original, the experimental design and data analysis of this study are reasonable, with clear logic and sufficient evidence, and the experimental results can support the conclusion of this paper.

The manuscript is well written, making it easily to follow. But I have the following comments and suggestions for the authors' consideration.

Response:

We sincerely appreciate the positive comments and comprehensive summary provided by the reviewer. Based on your constructive suggestions, we have made improvements in this revised manuscript.

(1) Due to the embryo lethality of the *Atmpk3 Atmpk6* double mutant, the authors used a chemically-rescued variant, *AtMPK3SR*, which mimics the double mutant in the presence of the inhibitor NA-PP1. How the authors ensure the dual-blocking effect is achieved after the application of NA-PP1.

Response:

We agree with the reviewer on this point. *AtMPK3SR* was an NA-PP1-sensitized MPK3 variant (MPK3^{TG})-rescued *mpk3 mpk6* double mutant plant (genotype: *mpk3 mpk6 P_{MPK3}:MPK3^{TA}*). The kinase activity of MPK3^{TG} can be specifically inhibited by NA-PP1 (Xu et al., 2014; Xu et al., 2016). According to the reviewer's comments, we analyzed the kinase activity of *AtMPK3* and *AtMPK6* in DMSO (as a control) or NA-PP1 treated- wild-type (Col-0) and *AtMPK3SR* after *S. sclerotiorum* inoculation. *S. sclerotiorum*-induced MPK3 and MPK6 activity was detected in both DMSO- and NA-PP1-treated Col-0. In DMSO-treated *AtMPK6SR*, only *S. sclerotiorum*-induced MPK3 activity was detected. However, in NA-PP1 treated *AtMPK6SR*, the *S. sclerotiorum*-induced MPK3 activity was completely inhibited. We added these findings to Supplementary Figure 12e, the Results section (lines 300-303) and the Methods section (lines 643-644 and 650) of the revised manuscript.

Newly added Supplementary Figure 12e

(e) Kinase activity of MPK3 and MPK6 in Col-0 and *AtMPK3SR* at 12 h after *S. sclerotiorum* inoculation. Col-0 and *AtMPK3SR* leaves were pretreated with DMSO or

NA-PP1 for 12 h before *S. sclerotiorum* inoculation. MPK3 and MPK6 activity were determined by immunoblot analysis using anti-pTEpY antibody. Molecular mass markers in kilodaltons are shown on the left.

References:

Xu, J. et al. A chemical genetic approach demonstrates that MPK3/MPK6 activation and NADPH oxidase-mediated oxidative burst are two independent signaling events in plant immunity. *Plant J.* **77**, 222-234 (2014).

Xu, J. et al. Pathogen-responsive MPK3 and MPK6 reprogram the biosynthesis of indole glucosinolates and their derivatives in Arabidopsis immunity. *Plant Cell* **28**, 1144-1162 (2016).

(2) The authors employed quadruple mutants of BnaMKK9, but the single mutant, particularly the BnaA07.MKK9 knock out mutants were not found in the manuscript.

Response:

Thank you very much for the comment. We regret that we did not obtain a *BnaA07.mkk9* single mutant in our study. Due to the high nucleotide and protein similarity among the four *BnaMKK9* homologs (Supplementary Figure 3a), we designed a CRISPR/Cas9 vector to target all four homologs simultaneously. Out of 11 T₀-positive transgenic plants, two contained simultaneous editing events at *BnaA07.MKK9*, *BnaC06.MKK9* and *BnaA02.MKK9* (Supplementary Table 2). After self-pollination, we obtained *Bnamkk9* quadruple mutants in T₄ progenies. Furthermore, three *BnaMKK9* homologs were significantly induced after *S. sclerotiorum* inoculation (Supplementary Figure 3b), indicating a potential redundancy in function among the *BnaMKK9* homologs. Consequently, we did not invest significant efforts in obtaining a single mutant.

(3) Why did the authors use three OE lines of BnaA07.MKK9 in the T₁ generation (Fig.1a-d), but only two lines in the T₂ generation (Supplementary Fig. 1)?

Response:

Thank you for this thoughtful comment. Both T₁ and T₂ generations of *BnaA07.MKK9^{DD}* OE lines were used for *S. sclerotiorum* resistance measurements. In T₁ generations, three *BnaA07.MKK9^{DD}* OE lines (#12, #35, and #51) were used. Unfortunately, constitutive overexpression of *BnaA07.MKK9^{DD}*, under control of the CaMV 35S promoter, resulted in several growth defects in oilseed rape. Notably, the *BnaA07.MKK9^{DD}* OE lines (#51) in T₁ generations exhibited severe growth defects and did not produce any seeds.

(4) The author did not conduct off-target analysis during gene editing (Fig 2e). How did they determined that the phenotype resulted from the edits of BnaMKK9?

Response:

Thank you for your insightful comment. A similar question was raised by Reviewer #1, (comment 3) Please refer to that section for our response.

(5) The gel images should be aligned in Fig.4g (the anti-GST).

Response:

Thank you for your careful review. We have redrawn Figure 4g in the revised manuscript.

(6) The author should provide clarification regarding the BnaA07.MKK9 haplotype of transgenic receptor J9712.

Response:

Thanks to the reviewer for these very important comments. The *BnaA07.MKK9* haplotype of transgenic receptor J9712 was Hap0. *BnaA07.MKK9*^{Hap1} and *BnaA07.MKK9*^{Hap2} were amplified from Niklas and SWU47. We added this information to the Methods section (lines 445, 488-490) in the revised manuscript.

(7) For qRT-PCR, the author utilized only one reference gene. It is recommended to employ at least two validated reference genes.

Response:

Thank you for this comment. In response, we included an additional reference gene for the qRT-PCR analysis. Specifically, *AtUBQ10* was used as the reference gene for *Arabidopsis*, and *BnUBC10* was used as the reference gene for *B. napus*. We added these findings to Supplementary Figure 6b, Supplementary Figure 11a, b, e, f, h, the Results section (lines 203, 287, 292 and 308), and the Methods section (lines 680). The specific primers were added into Supplementary Table 5. The qRT-PCR results remained consistent with the additional reference genes.

Newly added Supplementary Figure 6b

(d) Relative expression of *Csv::BnaA07.MKK9^{DD}* transgenic *Arabidopsis* lines (#1, #6, and #11) and Col-0 treated with either DMSO or 61.3 μ M methoxyfenozide (MOF). Samples were taken 8 h post-treatment, with *AtUBQ10* serving as the reference gene for expression analysis.

Newly added Supplementary Figure 11a, b, e, f, h

(a, b) Relative expression of *BnaC03.MPK3/BnaC03.MPK6* OE transgenic plants and J9712. (e, f) Relative expression of *BnaC03.MPK3/BnaC03.MPK6* OE transgenic plants in *Bnamkk9* knock out mutants. (h) Relative expression of *BnaA07.MKK9* in Col-0, *Csv::BnaA07.MKK9^{DD}*, *AtMPK3SR* and *Csv::BnaA07.MKK9^{DD}/AtMPK3SR* *Arabidopsis* plants. qRT-PCR was performed at 12 h after methoxyfenozide (MOF) and NA-PP1 treatment before *S. sclerotiorum* inoculation. In a, b, e and f *BnaUBC10* served as the reference gene. In h, *AtUBQ10* served as the reference gene.

(8) Figure 4a: Why used the kinase active variant, kinase inactive variant and wild type of *BnaA07.MKK9* to test the interaction of *BnaC03.MPK3* and *BnaC03.MPK6* by two-hybrid analysis? Only weak interaction was found between *BnaA07.MKK9KR* and *BnaC03.MPK3/6*. I recommend that the author provides possible explanations for this observation.

Response:

Thank you for this comment. The interaction between *BnaA07.MKK9* and *BnaC03.MPK3/6* could only be detected in the kinase inactive variant of *BnaA07.MKK9* by yeast two-hybrid analysis, but could be detected between kinase active variant and wild type of *BnaA07.MKK9* and *BnaC03.MPK3/6*. It appears that the active or wild type of *BnaA07.MKK9* may destabilize its interaction with *BnaC03.MPK3/6*. Similar results have been reported by Xu et al. (2018) and Cheng et al. (2015). We have added this information to the Results section (lines 244-245).

Reference:

Xu, R. et al. Control of grain size and weight by the OsMKKK10-OsMKK4-OsMAPK6 signaling pathway in rice. *Mol. Plant* **11**, 860-73 (2018).

Cheng, Z. et al. (2015). Pathogen-secreted proteases activate a novel plant immune pathway. *Nature* **521**, 213-216 (2015).

(9) Does the author conduct molecular detection for the Arabidopsis mutants before phenotypic analysis, such as *Atmkk9-1* (SALK_017378) and *Atmkk9-2* (SAIL_60_H06)?

Response:

Thank you for this insightful comment. The *Atmkk9-1* (SAIL_060_H06) and *Atmkk9-2* (SAIL_60_H06) mutants were confirmed with T-DNA border primers and gene-specific primers. The specific primers have been included in Supplementary Table 5. Additionally, the expression of *AtMKK9* in *Atmkk9-1* (SAIL_060_H06), *Atmkk9-2* (SAIL_60_H06) and Col-0 were also identified by RT-PCR (Supplementary Figure 4a, b). The results show that both *Atmkk9-1* and *Atmkk9-2* are homozygous knockout mutants.

(10) The numbers on the protein gel images lack units, kilodaltons?

Response:

Many thanks for pointing this out. The protein units were kilodaltons, and this information was added in to the Figure legends (lines 784, 860 and 898) and Supplementary Figure legends.

Responses to comments by Reviewer #3:

This paper reports cloning of a gene conferring what appears to be a strong resistance against *Sclerotinia sclerotiorum* in oilseed rape/rapeseed/canola (*Brassica napus*), one of the most important global oilseed crops. Stem rot caused by *Sclerotinia* is one of the most widespread, important and challenging diseases of rapeseed and canola, with very high importance in the three most important growing areas (China, Europe and North America). However, breeding for resistance remains challenging due to low heritability of resistance/susceptibility and a lack of knowledge about effective, quantitative resistance loci. *Sclerotinia* disease is also important in many of the world's other major oilseed, protein and vegetable crops (e.g. soybean, chickpea and other legumes, sunflower, potato and several vegetable species), however to date there has also been little progress in implementation of effective genetic resistance in other crops.

Hence, the present paper could represent an important breakthrough. I think the authors have done a very good job of identifying and characterising the resistance response activated by BnaA07.MKK9, so the work could become a catalyst for biotechnological approaches to raise resistance in *B. napus* and other crop species.

I am not an expert on plant-pathogen interactions or disease resistance mechanisms, hence I will limit my comments mainly to genetic aspects of the gene identification and validation, and the practical implications. Overall I think this is a clearly-formulated and coherent manuscript which sufficient value for publication. However I have major questions regarding the interpretation of the results and the practical relevance of this particular gene in oilseed rape breeding.

Response:

We are deeply grateful to the reviewer for recognizing the value of our work and for assisting us in improving the manuscript.

The GWAS was performed using phenotype data from 322 rapeseed accessions, Please add details in the first paragraph of the Methods section describing the rationale for assembly of this panel (particularly what is known about the diversity of the panel in relation to the species-wide diversity of *B. napus* which is reported elsewhere in the paper).

Response:

We apologize for not clearly describing the details of the GWAS panel. We have provided more details in the Methods section of the revised manuscript (Line 436-444).

The association panel used in this study consisted of 322 *B. napus* accessions, with a distribution of 286 accessions from Asia, 28 from Europe, 4 from North America, 3 from Oceania and 1 from South America (Supplementary Table 1). The selection of this association panel was deliberate, as the accessions showed a limited range of flowering times (days from sowing to flowering, Supplementary Table 1). This choice aimed to minimize the potential impact of developmental variations due to differences in flowering time among genotypes during the artificial *S. sclerotiorum* inoculation assay.

In accordance with the reviewer's suggestion, the genetic diversity and population structure of 322 accessions were analyzed. The results were provided in Supplementary Figure 1, the Results section Line 107-114 and Methods section Line 464-471 in the revised manuscript. The results revealed that the 322 accessions may be classified into two major sub-populations, designated as G1 and G2. (Supplementary Figure 1a). The G2 sub-population comprised 288 accessions; while the G1 sub-population comprised 34, with 26 of them being spring-type ecotypes (Supplementary Table 1). Principal component analysis (PCA) revealed that the G1 and G2 sub-populations formed distinct clusters, consistent with the findings from the structure analysis (Supplementary Fig 1b).

In particular, I am interested to know how the frequency of the resistance alleles in the GWAS panel reflects the bias found in the global *B. napus* panel, where the resistant haplotype Hap0 was subsequently found to be dominant in European winter and Asian semi-winter gene pools, though not so widespread in North American spring-type canola. To my knowledge no previous GWAS study of *Sclerotinia* resistance in a randomly assembled, genetically diverse population has revealed such a clear, major-effect QTL with high additive effects. Hence this result suggests that the GWAS panel has a very clear segregation for this major-gene effect that is not normally the case in (randomly selected) *B. napus* diversity panels.

Response:

We appreciate this reviewer's thoughtful comment. Approximately 49.1% (158/322), 19.2% (62/322) and 16.8% (54/322) of the accessions in the GWAS panel corresponded to haplotypes Hap0, Hap1, and Hap2, respectively (Response Fig. 1). This distribution differs slightly from the global *B. napus* panel (consisting of 2311 accessions in the BnIR database), where the frequencies of Hap0, Hap1, and Hap2 were 59.5% (1374/2311), 13.2% (304/2311), and 8.4% (194/2311), respectively (Response Fig. 1). The observed difference might be attributed to a slightly higher proportion of susceptible haplotypes (Hap1 and Hap2) in our GWAS panel. However, the overall distribution of haplotypes in the GWAS panel reflects that of the global panel (Response Fig. 1).

Furthermore, in the GWAS panel, there were 281, 35, and 6 accessions classified as semi-winter-type, spring-type, and winter-type, respectively. Similar to the global panel, the resistant haplotype Hap0 also exhibited dominance among semi-winter accessions at a rate of 63.1% (Hap1 14.6% and Hap2 22.2%), while the susceptible haplotype Hap1 was dominant in spring-type accessions with a frequency of 79.3% (Hap0 20.7% and Hap2 0%).

I agree with the reviewer's observation that despite several GWAS studies investigating SSR resistance in *B. napus* (Wei et al., 2016; Wu et al., 2016; Roy et al., 2021; 2022), none have identified major-effect QTLs with substantial additive effects. In contrast to these studies, our study differs in terms of population composition, marker density, and phenotypic evaluation methods, potentially influencing the outcomes of the GWAS

panel. We employed a total of 4,984,924 SNPs, generated from whole genome re-sequencing of this *B. napus* panel. The number of markers used in our study is 161-195 times greater than those in previous studies, which utilized 25,573 (Wu et al., 2016), 30,932 (Wei et al., 2016), and 25,809 (Roy et al., 2021; 2022) markers. Notably, Wu et al. (2016) and Wei et al. (2016) utilized the *Brassica napus* 60K Illumina Infinium™ SNP array. However, this array lacked marker coverage in the specific region of 500 Kb near the candidate gene, located at A07:24,504,013-24,504,939 in the 'Darmor-bzh' reference genome (refer to Supplementary Table 1 of Mason et al., 2017). Additionally, the method employed for assessing resistance to *S. sclerotiorum* differed from our study. We conducted stem inoculation at the mature stage in the field, aligning with the approach taken by Roy et al. (2021). This contrasts with the methods of Wu et al. (2016) and Wei et al. (2016), who utilized a detached stem inoculation assay, and also differs from Roy et al. (2022), who employed a petiole inoculation assay at the seedling stage.

Response Fig. 1 The frequency of haplotypes in the GWAS panel and the global *B. napus* panel

References:

- Mason, A. et al. A user guide to the *Brassica* 60K Illumina Infinium™ SNP genotyping array. *Theor. Appl. Genet.* **130**, 621-633 (2017).
- Roy, J. et al. Genome-wide association mapping and genomic prediction for adult stage sclerotinia stem rot resistance in *Brassica napus* (L) under field environments. *Sci. Rep.* **11**, 21773 (2021).
- Roy, J., del Río Mendoza, L. E., Bandillo, N., McClean, P. E. & Rahman, M. Genetic mapping and genomic prediction of sclerotinia stem rot resistance to rapeseed/canola (*Brassica napus* L.) at seedling stage. *Theor. Appl. Genet.* **135**, 2167-2184 (2022).
- Wei, L. et al. Genome-wide association analysis and differential expression analysis of resistance to Sclerotinia stem rot in *Brassica napus*. *Plant Biotechnol. J.* **14**, 1368-1380 (2016).
- Wu, J. et al. Genome-wide association study identifies new loci for resistance to Sclerotinia stem rot in *Brassica napus*. *Front. Plant Sci.* **7**, 1418 (2016).

Indeed, the data in Figure 6 show that susceptible haplotypes are rare in this species. Hence, one gains the impression that the paper is actually reporting cloning of a relatively rare susceptibility locus. In this case the results would have much less practical relevance for breeding than the manuscript suggests.

Response:

Thank you for voicing this important criticism. Indeed, the susceptible haplotypes (Hap1 and Hap2) are rare in winter-type accessions, a distinct pattern compared to semi-winter-type and spring-type accessions. Notably, approximately 30% of semi-winter and 72% of spring-type accessions lack the resistant haplotype (Hap0), suggesting that Hap0 could play a crucial role in enhancing SSR resistance specifically in semi-winter-type and spring-type *B. napus* varieties. At the very least, this locus can be used by breeders to screen whether their important breeding parents, elite lines, or germplasm resources contain disease-susceptible genes, which may greatly enhance the efficacy of disease-resistance breeding.

Figure 6 suggests that the resistance haplotype Hap0 is widespread in European winter and Asian semi-winter accessions, but less so in North American spring-type accessions. It further reports that the frequency of Hap0 was always very high, however it decreased in the period when global canola/oilseed rape breeding was focussed on conversion to 00 seed quality, and increased again in recent decades. What the authors do not show is how frequent Hap0 is in current spring-type, 00 canola cultivars in North America. I would suspect that the frequency is also very high, given that Canadian breeders have been selecting for quantitative *Sclerotinia* resistance for decades, so that a major-effect locus like this one would normally become quickly fixed for positive haplotypes through simple phenotypic selection. If this is not the case, the simplest explanation would be that the resistance is not effective against fungal pathotypes that are important in practical cropping systems in Canada.

Response:

Many thanks for your detailed and insightful comments. Nonetheless, we regret that we have not collected a sufficient number of current spring-type, 00 canola cultivars in North America for thorough investigation. We concur with the reviewer's suspicion that the frequency of Hap0 may progressively increase over time, especially considering the prolonged efforts of Canadian breeders who have been selecting for quantitative *Sclerotinia* resistance for several decades. We are also keenly interested in exploring this aspect further, and in our future research, we will acquire as many spring-type canola accessions from Canada as possible to conduct a comprehensive analysis.

How many of the Hap1 and Hap2 genotypes released since 2000 (right-hand diagram in Figure 6h) are North American/European spring-type cultivars, how many are European winter-type, and how many are Asian semi-winter type?

Response:

That is a very good question. We investigated the haplotypes of 418 *B. napus* accessions in different ecotypes and breeding eras, including 410 oil-use *B. napus* accessions, seven vegetable-use accessions and a synthetic *B. napus* accession, all of which were re-sequenced as detailed in the study by Hu et al. (2022). The 410 oil-use *B. napus* accessions were further categorized into spring (102), semi-winter (179) and winter (129) type accessions from China, Germany, France, Canada, Japan, the United States

and other countries (refer to Supplementary Table 1 of Hu et al., 2022). Notably, all accessions from the year 2000 onwards (spanning 2000-2011) exclusively belonged to the Asian semi-winter type. Among these accessions, 28, 3, and 6 were identified as Hap0, Hap1, and Hap2, respectively. Regrettably, at this point, we are unable to provide the specific data requested by the reviewers.

Reference:

Hu, J. et al. Genomic selection and genetic architecture of agronomic traits during modern rapeseed breeding. *Nat. Genet.* 54, 694-704 (2022).

In regard to practical relevance of the results, on line 422 the authors state that field experiments included two replications for the association panel, with 6-8 plants per accession being artificially inoculated on the stems with a single pathogen isolate (SS-1). Given that *Sclerotinia* infection typically has a very high environmental variance and G*E interaction (even with “controlled” stem inoculations, which are also not highly repeatable), I am concerned that there appears to be no validation in an independent field environment and/or with different isolates. Hence, it is not demonstrated that plants carrying the major resistance QTL against SS-1 are truly more resistant under natural *Sclerotinia* infection (including challenge by potentially multiple isolates) in different field environments. In the introduction, the authors emphasise the extreme damage caused by *Sclerotinia* in China and elsewhere (lines 48-52).

Response:

We agree that *Sclerotinia* infection typically demonstrates high environmental variance and G*E interaction. Due to the substantial workload for evaluating the sclerotinia stem rot (SSR) resistance of the entire set of accessions in the GWAS panel, we randomly selected 30 accessions from each haplotype within the GWAS panel to confirm the resistance performance in a distinct year/environment (Fig. 6c). The results showed that significant differences in SSR resistance were found among accessions with different haplotypes, with Hap0 accessions being the most resistant and those with Hap1 the least (Fig. 6c), suggesting that the effect of the QTL was stable across year/environment. Furthermore, in the evaluation of SSR resistance for all transgenic plants across different years and generations, consistent stability of the gene's effect was observed (Fig.2; Supplementary Figs. 2 and 3).

If the resistance is as effective as the manuscript suggests, how can the high frequency of resistance haplotype Hap0 in Chinese and European cultivars be reconciled with extremely high infection levels in farmer's fields?

Response:

Thank you for posing this excellent question. In our study, we identified a major locus for SSR resistance, and the advantageous haplotype of *BnaA07.MKK9* (Hap0) improved SSR resistance by ~30%. We agree with the reviewer's point: high infection levels are observed in the field despite the high frequency of pathogen resistant haplotype Hap0 in Chinese and European cultivars.. We speculate that the underlying reason may be the intricate genetic nature of SSR resistance in *B. napus*. This trait

appears to be governed by a highly complex genetic architecture influenced by multiple QTLs with additive effects. Resistance to necrotrophic pathogens is quantitative and requires many genes for resistance (Mengiste et al., 2012). Hence a single resistance locus or gene may not confer a high level of resistance. It will be important to clone multiple resistance genes, and aggregating various QTLs or candidate genes holds the potential to confer high SSR resistance. Furthermore, changes in climate variables (such as the warmer temperatures) may increase in severity and incidence of soil-borne fungal plant pathogens (Delgado-Baquerizo et al., 2020).

References:

Mengiste, T. Plant immunity to necrotrophs. *Annu. Rev. Phytopathol.* **50**, 267-294 (2012).

Delgado-Baquerizo, M. et al. The proportion of soil-borne pathogens increases with warming at the global scale. *Nat. Clim. Change.* **10**, 550-554 (2020).

I also find it unusual that a leaf assay was used alongside stem inoculations to screen resistance in transgenic plants, but not in the association panel. To my knowledge and from personal experience, the transferability of resistance/susceptibility scores between leaf and stem assays is generally very poor (even if it may correlate in specific, extreme genotypes with specific isolates). Indeed, Brassica pathologists are still very divided about which (if any) of these methods gives truly relevant results for natural *Sclerotinia* infections in a field cropping situation. Most agree that an isolate mixture (rather than single isolates) come closer to a natural infection situation, although individual isolates are important for detailed characterisation of plant responses. However, in the transgenic plants there seems to be a high convergence of the disease scores. If this simple assay discriminates so well for this gene, is there a plausible reason why the authors did not support their stem inoculation data in the GWAS with additional leaf inoculation data? Also, why were Hap0-carrying accessions not challenged with other pathogen isolates?

Response:

We appreciate this candid comment. In our previous study, we assessed the SSR resistance of the HJ-DH population, consisting of 190 individual DH lines, using both detached leaf inoculation and stem inoculation. Although a significant positive correlation between leaf resistance and stem resistance was observed across most environments (Wu et al., 2013), the correlation coefficients ranged only from 0.18 to 0.46 ($P < 0.01$). Hence, we agree with the reviewer's suggestion that there may be an inconsistency between stem and leaf resistance. Given that stem rot at the mature plant stage is the primary cause of yield loss post-infection by *S. sclerotiorum* in oilseed rape, our primary focus on *S. sclerotiorum* resistance is to identify QTLs associated with stem resistance, primarily using stem inoculation for resistance assessment. Consequently, we did not perform leaf inoculation for the GWAS panel.

In order to quickly assess the resistance of transgenic plants, we used leaf inoculation during the seedling stage. We found that the candidate gene also influences leaf resistance. Consequently, in subsequent experiments involving the inoculation of

transgenic lines, we employed both leaf inoculation and stem inoculation methods simultaneously.

Following the reviewer's recommendation, we selected 20 accessions for each haplotype from the material we've cultivated in the field to validate leaf resistance performance at the seeding stage. Consistent with the differences observed in stem resistance, significant differences in leaf resistance were likewise found among accessions with these haplotypes (Line 322-325 Supplementary Figure 13 in the revised manuscript).

Newly added Supplementary Figure 13

Sclerotinia stem rot (SSR) resistance of 20 randomly selected accessions for each haplotype evaluated by detached-leaf inoculations. The boxes show the median and the lower and upper quartiles. Different lowercase letters indicate significant differences ($P < 0.05$) determined with one-way analysis of variance (ANOVA) and Tukey's honestly significant difference (HSD) test.

Effector-triggered immunity (ETI), which confers resistance to biotrophic and hemibiotrophic pathogens, conforms to the gene-for-gene hypothesis, as it is dominated by direct and indirect interactions between resistance genes (R gene) and Avr effectors. For this type of pathogen, different physiological races may possess distinct Avr effectors. Therefore, it is common practice to investigate the effectiveness of R genes by using multiple physiological races. Conversely, resistance to necrotrophic pathogens is largely independent of ETI. Instead, immune responses triggered by diverse pathogen-associated molecular patterns (PAMPs) confer quantitative resistance to broad host necrotrophic pathogens, referred to as PAMP-triggered immunity (PTI) (Liao et al., 2022). PTI does not typically exhibit specificity toward particular pathogen strains or races. Most published literature on resistance to *Botrytis cinerea* has also focused on a single isolate, B05.10 (Wang et al., 2015; Berriri et al., 2016; Du et al., 2017). Consequently, we didn't consider conducting experiments with multiple isolates of *S. sclerotiorum*. The isolate used in this study, SS-1, exhibits aggressive virulence. It was generously provided by Prof. Guoqing Li from the State Key Laboratory of Agricultural Microbiology, Huazhong agricultural University. We have consistently

employed this particular isolate in our research for over 15 years. Based on the reviewer's comments, in our future studies, we will isolate and purify additional strains to assess resistance of *B. napus*. Simultaneously, we will place greater emphasis on natural disease occurrences, as these instances often involve mixed isolates.

References:

Berriri, S. et al. SWR1 chromatin-remodeling complex subunits and H2A.Z have non-overlapping functions in immunity and gene regulation in *Arabidopsis*. *Mol. Plant* **9**, 1051-1065 (2016).

Du, M. et al. MYC2 orchestrates a hierarchical transcriptional cascade that regulates jasmonate-mediated plant immunity in tomato. *Plant Cell* **29**, 1883-1906 (2017).

Liao, C-J. et al. Pathogenic strategies and immune mechanisms to necrotrophs: Differences and similarities to biotrophs and hemibiotrophs. *Curr. Opin. Plant Biol.* **69** 102291 (2022).

Wang, C. et al. Arabidopsis Elongator subunit 2 positively contributes to resistance to the necrotrophic fungal pathogens *Botrytis cinerea* and *Alternaria brassicicola*. *Plant J.* **83**,1019-1033 (2015).

Wu, J. et al. Identification of QTLs for resistance to sclerotinia stem rot and *BnaC.IGMT5.a* as a candidate gene of the major resistant QTL *SRC6* in *Brassica napus*. *PloS one* **8**, e67740 (2013).

Furthermore, no information is provided on the heritability of the infection response, which is generally very low for Sclerotinia disease in field situations. Since plants for the GWAS were grown in only a single environment, the authors can only calculate h^2 across the two replications (i.e. “repeatability”) as a proxy for heritability. At least that might improve confidence in the reliability of the phenotype data (with this single isolate) in the absence of multiple field environments.

Response:

Thank you for your insightful comments. In accordance with the reviewer's suggestion, we conducted the ANOVA analysis (Response Table 1) and calculated the heritability using the formula $h^2 = V_G/(V_G + V_E)$, where V_G is the genetic component and V_E is the error variance. The estimated heritability (h^2) for SSR resistance is 72%, which is consistent with previous studies that have reported the heritability for SSR resistance within the range of 61.7-88% (Wei et al., 2016; Wu et al., 2016; Roy et al., 2021; 2022). Furthermore, a significant positive correlation ($r = 0.73$, $P < 0.01$) between the two replications was observed, suggesting that the phenotype data in this study is reliable. The results were provided in Line 102-104 in the revised manuscript.

Response Table 1. Analysis of variance (ANOVA) of Sclerotinia stem rot resistance trait in *B. napus*

	df	Mean square	F value	P value
Replication	1	2.28	0.47	0.49
Genotype	321	8.25	6.16	<.0001

References

- Roy, J. et al. Genome-wide association mapping and genomic prediction for adult stage sclerotinia stem rot resistance in *Brassica napus* (L) under field environments. *Sci. Rep.* **11**, 21773 (2021).
- Roy, J., del Río Mendoza, L. E., Bandillo, N., McClean, P. E. & Rahman, M. Genetic mapping and genomic prediction of sclerotinia stem rot resistance to rapeseed/canola (*Brassica napus* L.) at seedling stage. *Theor. Appl. Genet.* **135**, 2167-2184 (2022).
- Wei, L. et al. Genome-wide association analysis and differential expression analysis of resistance to Sclerotinia stem rot in *Brassica napus*. *Plant Biotechnol. J.* **14**, 1368-1380 (2016).
- Wu, J. et al. Genome-wide association study identifies new loci for resistance to Sclerotinia stem rot in *Brassica napus*. *Front. Plant Sci.* **7**, 1418 (2016).

Overall, I think the manuscript reports cloning of an interesting susceptibility gene to a major disease in a major crop, which is a novel finding. However, the resistant haplotype of this gene is already widespread in major oilseed rape/canola cultivars (and I would be rather surprised if it is not more or less fixed in the most recent, high-yielding 00-type cultivars in all ecotype groups). Hence, more detailed results and explanations are needed to convince me that the gene has a real practical relevance for breeding that has not already been exploited by simple phenotypic selection for disease scores and yield performance in modern cultivars. This is a solid and well-written manuscript but in light of these concerns I consider the main message and conclusions to be somewhat exaggerated.

Response:

We greatly appreciate the reviewer's constructive comments of our manuscript. Regarding the concerns raised by the reviewers, we have provided some explanations in the responses above. Despite the advantageous haplotype of *BnaA07.MKK9* already being widespread in major oilseed rape cultivars, many cultivars still lack this advantageous haplotype. At the very least, we can eliminate whether the crucial breeding materials at hand contain the susceptible haplotype. Consequently, this gene retains substantial practical significance. Moreover, our findings unveil a previously unknown role of the MKK9-MPK3/6 cascade in conferring resistant to *S. sclerotiorum*, enriching our understanding of plant defense mechanisms against necrotrophic pathogens. A comprehensive understanding of the molecular mechanisms governing host-pathogen interactions should assume a pivotal role in guiding resistance breeding.

Lines 33-34: "Rapeseed (*Brassica napus*)" – I recommend to refer throughout the manuscript to "oilseed rape" rather than rapeseed. This is necessary to distinguish 00-quality oilseed rape/canola types (the form that is today grown globally) from old ++ or +0 types which are barely grown today (internationally commonly referred to as rapeseed in English). To emphasise the importance of the findings for all common B.

napus crop types, I recommend to change this particular phrase to “Oilseed rape (Brassica napus), also known as canola or rapeseed, is a globally important edible oil crop that emerged...”

Response:

Many thanks for this comment. We have changed “rapeseed” to “oilseed rape” throughout the revised manuscript.

Lines 33-34: “...around 7,500 years ago ...”

The supposedly exact timing of the origin of *B. napus* 7,500 years ago is rather controversial and I personally prefer to avoid using this explicit date. In the genome assembly paper of Chalhoub et al. (2014) in *Science*, this wording was introduced in the abstract of the manuscript by the journal editors, although the authors actually stressed that the evidence from the “molecular clock” data placed the origin of *B. napus* at “no older than 7,500 years”. As a consequence, this date is frequently misconstrued as fact: However if you read Chalhoub et al. (2014) carefully, the main manuscript text states that *B. napus* formed after this date. As noted in the present manuscript on line 37, the first indications for *B. napus* cultivation arose in the 13th Century and there is no evidence that the species arose earlier. Since no natural forms have ever been found, the available evidence suggests that the time of origin was considerably more recent than the “maximum” 7,500 years ago. I would prefer if you change your statement to “a globally important edible oil crop that emerged during the past 7,500 years”.

Response:

Many thanks for this helpful suggestion. According to the review’s comments, we have revised the sentences accordingly (Line 33-36 in the revised manuscript).

Line 34: Please change “from a natural hybridization event” to “from natural hybridisation”. To my knowledge it is unclear whether there was one or several hybridisation events, but the latter appears more likely.

Response:

We agree with the reviewer’s comment and changed “from a natural hybridization event” to “from natural hybridization” (Line 35) in the revised manuscript.

Lines 41-44: “Double low breeding technique” is not a technique for breeding, rather double low cultivars are simply the result of classical breeding implementing chemical analysis for seed quality selection. Please change to “Over the past five decades, double-low cultivars with low seed erucic acid and glucosinolate content, pioneered in Canada, significantly improved *B. napus* oil and meal quality.”

Response:

Thank you very much for your careful review. We have modified the sentences in the revised manuscript (Line 41-43).

Line 83: “we identified “a particular gene”, BnaA07.MKK9, as a significant factor in

SSR resistance in rapeseed. Please change to “we identified the MKK gene BnaA07.MKK9 as a significant factor contributing to SSR resistance in rapeseed.

Response:

Many thanks for this constructive suggestion. According to the review’s comments, we have revised the sentences accordingly (Line 85-86 in the revised manuscript).

REVIEWER COMMENTS

Reviewer #1 (Remarks to the Author):

The authors have addressed all my concerns, and they have truly performed an incredibly work in the field of Sclerotinia stem rot resistance, so I would like to recommend the paper for publication.

Reviewer #2 (Remarks to the Author):

I have read the revised manuscript provided by the authors and the replies to all reviewers' opinions in detail. I think the Point-by-point response of the author to the opinions and suggestions of the three reviewers is detailed, clear and candid. In the revised manuscript, the author made careful revisions according to the problems found by the reviewers, modified and supplemented some unclear places in the original manuscript, and added some necessary charts and information to make the article clearer and more logical. So I think the revised manuscript has reached the publication level.

Reviewer #3 (Remarks to the Author):

The authors have responded carefully to the concerns raised in my previous review and in many cases were able to alleviate these concerns or make appropriate corrections or additions, e.g. regarding heritability. In principle I still believe the manuscript provides interesting new information which is worthwhile to publish and gives nice insight into resistance mechanisms of a major pathogen in a major global crop. However, no additional data is presented to overcome my strong suspicion that the variant conferring resistance (Hap 0) is already very widespread in current cultivars and probably does not have a particularly high relevance for modern breeding.

Notably, the authors state that all investigated accessions released from the year 2000 onwards (but in fact only spanning the period from 2000-2011) belonged exclusively to the Asian semi-winter ecotype, i.e. no recent European winter-type or North American spring-type cultivars were investigated in the study. Also, the revision provides no new data on haplotype frequencies in recent winter/semi-winter cultivars. According to Figure 6, 93% of the tested winter-type cultivars (all pre-2000) contain the resistance haplotype. Given the major breeding progress in Europe in the

past 23 years, I would be very surprised if a haplotype with such a strong heritability and effect had not meanwhile been completely fixed in modern, European winter-type hybrid cultivars. Similarly, in North America (where breeding progress is fast due to having two cycles per year), the authors cannot discount a similar strong increase in Hap 0 frequency in Canadian spring-type cultivars unless they provide data from modern cultivars released in the the past 23 years. Even for the Asian semi-winter cultivars, no data is presented from cultivars released in the past 13 years (these are what I would call "modern", pre-2011 inbred cultivars are no longer competitive with current oilseed rape hybrids).

Given this shortcoming, I reiterate my assumption that the resistant haplotype is likely already widespread in modern oilseed rape cultivars. Unless the authors are able to provide evidence which suggests otherwise (i.e. data showing that most major current cultivars of the different ecotypes, released during the past 5-10 years, do not already contain the resistance haplotype), I strongly recommend to modify the language accordingly and refrain from assumptions that the information can have a "significant impact in breeding". In fact, as the authors acknowledge, this resistance locus already did have a very strong impact in the previous century (even without knowledge of the underlying gene), and that impact has most likely continued throughout the past quarter of a century (without knowledge of the underlying gene). Nevertheless, this resistance alone has by no means the problem of Sclerotinia disease in modern rapeseed crops.

Nevertheless, the identification of a gene with a notable impact on Sclerotinia infection is an important finding in general for crop production, since this is an important disease in several oilseed and protein crops. Knowledge of the genetics underlying resistance mechanisms may help engineer novel diversity for future breeding. I believe this message would be more appropriate than the present conclusions in regard to BnaA07.MKK9 as a "cornerstone" for "informed breeding of oilseed rape".

Reviewer #4 (Remarks to the Author):

The work presented here is a resubmission that concerns the resistance of oilseed rape (*Brassica napus*) against sclerotinia stem rot, caused by the necrotrophic fungus *Sclerotinia sclerotiorum*. By performing a genome-wide association study, the authors have identified an important regulator of sclerotinia resistance in oilseed rape; BnaA07.MKK9, which is a mitogen-activated protein kinase-kinase (MAPKK) that forms part of a MAPKKK module and phosphorylates downstream MPK3 and MPK6 proteins of this module. The authors show that the BnaA07.MKK9 protein is a key regulator of resistance of oilseed rape to the fungus and that natural variation in the encoding gene forms an important genetic resource for breeding for resistance of oilseed rape.

The finding that MAPK cascades play a role in resistance of plants to pathogens is not new. There are numerous examples of already more than 30 years ago showing that these MAPK modules play

an essential role in resistance of plants to various pathogens and pests. These MAPKs are important regulators, but are not resistance (R) genes or susceptibility (S) genes and I agree with Reviewer 3 that “..... in light of these concerns I consider the main message and conclusions to be somewhat exaggerated.”

The authors responded very well to the concerns of the reviewers and have done a lot of impressive work. The finding that the BnaA07.MKK9 protein is a key regulator of resistance of oilseed rape to *S. sclerotiorum* is very interesting, but not new. The demonstration that natural variation in the encoding gene forms an important genetic resource for breeding for resistance of oilseed rape is new, but it could well be possible that this trait is already being used as a result of selection, and this should be checked for all currently employed more resistant cultivars of oilseed rape.

Point-by-point response to the reviewers' comments

Responses to comments by Reviewer #1:

The authors have addressed all my concerns, and they have truly performed an incredibly work in the field of Sclerotinia stem rot resistance, so I would like to recommend the paper for publication.

Response:

We sincerely express our gratitude to the reviewer for your insightful comments on our manuscript. Your critical and invaluable feedback has significantly enhanced the quality of our work. We are truly appreciative of the time and effort you have dedicated to reviewing our manuscript.

Responses to comments by Reviewer #2:

I have read the revised manuscript provided by the authors and the replies to all reviewers' opinions in detail. I think the Point-by-point response of the author to the opinions and suggestions of the three reviewers is detailed, clear and candid. In the revised manuscript, the author made careful revisions according to the problems found by the reviewers, modified and supplemented some unclear places in the original manuscript, and added some necessary charts and information to make the article clearer and more logical. So I think the revised manuscript has reached the publication level.

Response:

We sincerely thank the reviewer for your thoughtful comments on our manuscript. Your positive feedback is greatly appreciated. We appreciate the effort you put into reviewing our manuscript.

Responses to comments by Reviewer #3:

1. The authors have responded carefully to the concerns raised in my previous review and in many cases were able to alleviate these concerns or make appropriate corrections or additions, e.g. regarding heritability. In principle I still believe the manuscript provides interesting new information which is worthwhile to publish and gives nice insight into resistance mechanisms of a major pathogen in a major global crop.

Response:

We are deeply grateful to the reviewer for your time and expertise in reviewing our manuscript. Thank you for recognizing the value of our work and for assisting us in improving the manuscript.

2. However, no additional data is presented to overcome my strong suspicion that the variant conferring resistance (Hap 0) is already very widespread in current cultivars and probably does not have a particularly high relevance for modern breeding.

Notably, the authors state that all investigated accessions released from the year 2000

onwards (but in fact only spanning the period from 2000-2011) belonged exclusively to the Asian semi-winter ecotype, i.e. no recent European winter-type or North American spring-type cultivars were investigated in the study. Also, the revision provides no new data on haplotype frequencies in recent winter/semi-winter cultivars. According to Figure 6, 93% of the tested winter-type cultivars (all pre-2000) contain the resistance haplotype. Given the major breeding progress in Europe in the past 23 years, I would be very surprised if a haplotype with such a strong heritability and effect had not meanwhile been completely fixed in modern, European winter-type hybrid cultivars. Similarly, in North America (where breeding progress is fast due to having two cycles per year), the authors cannot discount a similar strong increase in Hap 0 frequency in Canadian spring-type cultivars unless they provide data from modern cultivars released in the the past 23 years. Even for the Asian semi-winter cultivars, no data is presented from cultivars released in the past 13 years (these are what I would call "modern", pre-2011 inbred cultivars are no longer competitive with current oilseed rape hybrids).

Given this shortcoming, I reiterate my assumption that the resistant haplotype is likely already widespread in modern oilseed rape cultivars. Unless the authors are able to provide evidence which suggests otherwise (i.e. data showing that most major current cultivars of the different ecotypes, released during the past 5-10 years, do not already contain the resistance haplotype), I strongly recommend to modify the language accordingly and refrain from assumptions that the information can have a "significant impact in breeding". In fact, as the authors acknowledge, this resistance locus already did have a very strong impact in the previous century (even without knowledge of the underlying gene), and that impact has most likely continued throughout the past quarter of a century (without knowledge of the underlying gene). Nevertheless, this resistance alone has by no means the problem of Sclerotinia disease in modern rapeseed crops.

Nevertheless, the identification of a gene with a notable impact on Sclerotinia infection is an important finding in general for crop production, since this is an important disease in several oilseed and protein crops. Knowledge of the genetics underlying resistance mechanisms may help engineer novel diversity for future breeding. I believe this message would be more appropriate than the present conclusions in regard to BnaA07.MKK9 as a "cornerstone" for "informed breeding of oilseed rape".

Response:

Many thanks for your detailed and insightful comments. Additionally, we truly value the reviewer's encouraging comment, stating that “the identification of a gene with a notable impact on Sclerotinia infection is an important finding in general for crop production”.

We regret that we have not been able to gather a sufficient amount of current oilseed rape cultivars encompassing different ecotypes, which have been released globally in the past 5-10 years. Consequently, at this point, we are unable to provide the specific

data about the distribution of the resistant haplotype Hap0 in current oilseed rape cultivars. So we concur with the reviewer's suggestion, and have accordingly modified our language to reflect this limitation. The specific changes are outlined below:

Line 29-31:

Original: These findings enhance our understanding of SSR resistance and offer a valuable genetic resource for the informed breeding of oilseed rape.

Modified: These findings enhance our understanding of SSR resistance and may help engineer novel diversity for future breeding of oilseed rape.

Line 93-94:

Original: This discovery highlights the importance of *BnaA07.MKK9* in SSR resistance and provides a valuable allele for breeding SSR resistant oilseed rape varieties.

Modified: This discovery highlights the importance of *BnaA07.MKK9* in SSR resistance and enhances our understanding of the genetic and molecular basis underlying SSR resistance in oilseed rape.

Line 431-433

Original: We propose that *BnaA07.MKK9* should be a cornerstone for breeding oilseed rape.

Modified: These findings provide valuable insights into the genetic and molecular mechanisms underlying SSR resistance, thereby holding the potential to facilitate the engineering of novel diversity for future breeding in oilseed rape.

Responses to comments by Reviewer #4:

1. The work presented here is a resubmission that concerns the resistance of oilseed rape (*Brassica napus*) against sclerotinia stem rot, caused by the necrotrophic fungus *Sclerotinia sclerotiorum*. By performing a genome-wide association study, the authors have identified an important regulator of sclerotinia resistance in oilseed rape; *BnaA07.MKK9*, which is a mitogen-activated protein kinase-kinase (MAPKK) that forms part of a MAPKKK module and phosphorylates downstream MPK3 and MPK6 proteins of this module. The authors show that the *BnaA07.MKK9* protein is a key regulator of resistance of oilseed rape to the fungus and that natural variation in the encoding gene forms an important genetic resource for breeding for resistance of oilseed rape.

The finding that MAPK cascades play a role in resistance of plants to pathogens is not new. There are numerous examples of already more than 30 years ago showing that these MAPK modules play an essential role in resistance of plants to various pathogens and pests. These MAPKs are important regulators, but are not resistance (R) genes or susceptibility (S) genes and I agree with Reviewer 3 that “..... in light of these concerns I consider the main message and conclusions to be somewhat exaggerated.”

Response:

Thank you for your invaluable comments and suggestions regarding our manuscript. Sclerotinia stem rot (SSR), caused by the necrotrophic fungus *Sclerotinia sclerotiorum*, stands as one of the most devastating diseases affecting several oilseed and protein crops, including oilseed rape, soybean, and sunflower. Previous studies on these crops have demonstrated that SSR resistance is a highly complex quantitative trait governed by multiple quantitative trait loci (QTL). Despite extensive research, no QTL has been successfully cloned, leading to confusion regarding the mechanisms underlying this quantitative resistance. Our study has made a breakthrough by identifying *BnaA07.MKK9* as a key regulator of SSR resistance in oilseed rape through genome-wide association study. The successful cloning of this gene represents a significant step forward in our comprehension of plant-necrotrophic pathogen interactions and offers new insights and valuable information for future research in this area.

We fully agree with the reviewer's perspective that *BnaA07.MKK9* is not an *R* gene but rather serves as a key regulator of SSR resistance. Indeed, apart from the *Arabidopsis RLM3* (a TIR domain encoding gene), there are no report of any single resistance (*R*) gene that can provide dominant resistance against necrotrophic fungi (Staal et al., 2008; Ghozlan et al., 2020; Gaona et al., 2023). Resistance to necrotrophs has been found to be partial (Ghozlan et al., 2020; Gaona et al., 2023). For broad host-range necrotrophic fungi like *Botrytis cinerea* and *S. sclerotiorum*, immune responses triggered by diverse PAMPs (PAMP-triggered immunity, PTI) confer quantitative resistance to these pathogens (Liao et al., 2022). As mentioned in the Introduction, PTI, a form of quantitative resistance, provides broader protection and is vital in the defense against necrotrophic pathogens. Therefore, the identification of critical genetic regulators mediating resistance to necrotrophic fungi holds significant importance. The aggregation of various elite regulator genes involved in PTI has the potential to confer high resistance to these pathogens. So, even though *BnaA07.MKK9* is not an *R* gene, our results provide valuable insights into the genetic and molecular mechanisms underlying SSR resistance.

We have revised the language and refrained from making any assumptions regarding the "significant impact in breeding," as suggested by Reviewers 3 and 4. Please refer to our response to Reviewer 3.

2. The authors responded very well to the concerns of the reviewers and have done a lot of impressive work. The finding that the *BnaA07.MKK9* protein is a key regulator of resistance of oilseed rape to *S. sclerotiorum* is very interesting, but not new.

Response:

We deeply appreciate the reviewer's constructive feedback and acknowledgment of the value of our findings. As the reviewer pointed out, there are numerous examples of MAPK modules that play an essential role in resistance of plants to various pathogens and pests. Most of these findings were described in model plants using reverse genetics.

For example, two *Arabidopsis* MAPK cascades, MEKK1–MKK1/2–MPK4/MPK11 and MKKK3/5–MKK4/5–MPK3/6, are known to contribute to plant immunity (Zhang et al., 2022). Although MAPK is known to be involved in plant disease resistance, we do not yet know how to effectively utilize MAPK cascades to improve their value in crop disease resistance breeding. We found that the different kinase activities of the *BnaA07.MKK9* haplotypes affect the timing and strength of the disease resistance response, leading to different levels of resistance to *S. sclerotiorum*. This discovery offers new insights: the natural variation within the MAPK gene represents a valuable genetic resource for crop resistance breeding.

Moreover, among the ten MKK genes, the roles of *MKK1/2* (Group A MKKs) and *MKK4/5* (Group C MKKs) in plant immunity have been well studied (Bi et al., 2018; Sun et al., 2018; Yan et al., 2018; Yamada et al., 2016). Little is known about Group D MKK genes, including *MKK7*, *MKK8*, *MKK9*, and *MKK10*, in plant immunity. Our findings unveil a previously unknown role of the MKK9 (Group D MKK) and the MKK9-MPK3/6 cascade in conferring resistance to pathogens. This enriches our understanding of the MAPK cascade in plant immune signaling, particularly in the context of plant defense mechanisms against necrotrophic pathogens.

Hence, our finding provides new information and valuable insights into resistance mechanisms against a significant pathogen in a globally important crop.

References:

- Bi, G. et al. Receptor-like cytoplasmic kinases directly link diverse pattern recognition receptors to the activation of mitogen-activated protein kinase cascades in *Arabidopsis*. *Plant Cell* **30**, 1543–1561 (2018).
- Liao, C. J. et al. Pathogenic strategies and immune mechanisms to necrotrophs: Differences and similarities to biotrophs and hemibiotrophs. *Curr. Opin. Plant Biol.* **69**, 102291 (2022).
- Gaona, M. R. et al. Mutation of PUB17 in tomato leads to reduced susceptibility to necrotrophic fungi. *Plant Biotechnol. J.* **21**, 2157 (2023).
- Ghozlan, M. H. et al. Plant defense against necrotrophic pathogens. *American J. Plant Sci.* **11**, 2122–2138 (2020).
- Sun, T. et al. Antagonistic interactions between two MAP kinase cascades in plant development and immune signaling. *EMBO Rep.* **19**, e45324 (2018).
- Staal, J. et al. RLM3, a TIR domain encoding gene involved in broad-range immunity of *Arabidopsis* to necrotrophic fungal pathogens. *Plant J.* **55**, 188–200 (2008).
- Yamada, K. et al. The *Arabidopsis* CERK1-associated kinase PBL27 connects chitin perception to MAPK activation. *EMBO J.* **35**, 2468–2483 (2016).
- Yan, H. et al. BRASSINOSTEROID-SIGNALING KINASE1 phosphorylates MAPKKK5 to regulate immunity in *Arabidopsis*. *Plant Physiol.* **176**, 2991–3002 (2018).

3. The demonstration that natural variation in the encoding gene forms an important

genetic resource for breeding for resistance of oilseed rape is new, but it could well be possible that this trait is already being used as a result of selection, and this should be checked for all currently employed more resistant cultivars of oilseed rape.

Response:

Thank you for your insightful comment. A similar question was raised by Reviewer 3. Please refer to that section for our response.

REVIEWERS' COMMENTS

Reviewer #3 (Remarks to the Author):

It is disappointing that the authors were not able to investigate allele frequencies of the putative "resistance" locus in recent cultivars, as I am quite sure they would have found that a major gene variant causing high susceptibility of old cultivars to *Sclerotinia* would have already been more or less completely eliminated in modern cultivars of all major rapeseed/canola ecotypes through simple field selection, given the high levels of *Sclerotinia* disease in major production areas and especially China (as stressed by the authors themselves).

In this context, I understand that it can be difficult for researchers in China to obtain seeds from current North American or European hybrid cultivars. However, it does seem unusual that the authors were unable to obtain any cultivars released during the past 13 years in China. Without data from recent cultivars, one can only speculate on what impact the results might have for rapeseed/canola breeding in China or elsewhere, but unfortunately I strongly suspect that the impact will not be high.

With this in mind, it is pleasing to see that the authors at least agreed to change the wording in the manuscript regarding the impact of their results for breeding. I will let the editor decide whether or not the manuscript still has sufficient scope or impact for a Nature journal without data representing the current situation in rapeseed/canola production.

If accepted for publication, I recommend the following minor change in the abstract: "Furthermore, variations in the coding sequence of BnaA07.MKK9 altered its kinase activity and improved SSR resistance by ~30% in CULTIVARS CARRYING the advantageous haplotype".

Reviewer #4 (Remarks to the Author):

Again, the authors responded very well to the concerns of the reviewers. The finding that the BnaA07.MKK9 protein is a key regulator of resistance of oilseed rape to *S. sclerotiorum* is very interesting, but not new. I remain with my remark (similar to Reviewer 3) that it could well be possible that this trait is already being used as a result of selection. However, given the fact that this concerns an important disease in an important crop plant, this work is still of high relevance.

Point-by-point response to the reviewers' comments

Responses to comments by Reviewer #3:

It is disappointing that the authors were not able to investigate allele frequencies of the putative "resistance" locus in recent cultivars, as I am quite sure they would have found that a major gene variant causing high susceptibility of old cultivars to Sclerotinia would have already been more or less completely eliminated in modern cultivars of all major rapeseed/canola ecotypes through simple field selection, given the high levels of Sclerotinia disease in major production areas and especially China (as stressed by the authors themselves).

In this context, I understand that it can be difficult for researchers in China to obtain seeds from current North American or European hybrid cultivars. However, it does seem unusual that the authors were unable to obtain any cultivars released during the past 13 years in China. Without data from recent cultivars, one can only speculate on what impact the results might have for rapeseed/canola breeding in China or elsewhere, but unfortunately I strongly suspect that the impact will not be high.

With this in mind, it is pleasing to see that the authors at least agreed to change the wording in the manuscript regarding the impact of their results for breeding. I will let the editor decide whether or not the manuscript still has sufficient scope or impact for a Nature journal without data representing the current situation in rapeseed/canola production.

If accepted for publication, I recommend the following minor change in the abstract: "Furthermore, variations in the coding sequence of BnaA07.MKK9 altered its kinase activity and improved SSR resistance by ~30% in CULTIVARS CARRYING the advantageous haplotype". .

Response:

We sincerely express our gratitude to the reviewer for your insightful comments on our manuscript. As per the reviewer's suggestion, we have made the minor change to the abstract (line 29).

Responses to comments by Reviewer #4:

Again, the authors responded very well to the concerns of the reviewers. The finding that the BnaA07.MKK9 protein is a key regulator of resistance of oilseed rape to *S. sclerotiorum* is very interesting, but not new. I remain with my remark (similar to Reviewer 3) that it could well be possible that this trait is already being used as a result of selection. However, given the fact that this concerns an important disease in an important crop plant, this work is still of high relevance.

Response:

We deeply appreciate the reviewer's recognition of the value in our manuscript and the thoughtful, insightful comments you have provided.